# Deciphering DEL pocket patterns through contrastive learning

Wenyi Zhang [1,2], Yuxing Wang [1,2], Rui Zhan [1,2], Runtong Qian [1,2], Qi Hu [1,2] & Jing Huang [1,2] ✉

DNA-encoded libraries (DELs) facilitate high-throughput screening of trillions of molecules against protein targets through split-pool synthesis and DNA tagging. Despite their potential, only a few DEL-derived compounds have advanced to clinical trials or reached the market. A better understanding of the defining characteristics of target proteins, particularly those with binding pockets suitable for DEL screening, is critical to improving success rates. However, existing approaches remain limited in assessing pocket flexibility and functional similarity. Here, we present ErePOC, a pocket representation model based on contrastive learning with ESM-2 embeddings to address these challenges. ErePOC captures both structural and functional features of binding pockets, enabling identification of shared characteristics among DEL targets. By integrating analyses of low-dimensional physicochemical properties and high-dimensional ErePOC embeddings, we provide a comprehensive view of DEL target space. With 98% precision in downstream classification tasks, ErePOC demonstrates high performance in pocket representation, which is then applied to predict human proteins suitable for DEL screening, with enrichment uncovered across 18 functional categories. This work establishes a framework for enhancing DEL-based drug discovery through more effective target selection and pocket similarity analysis.

DNA-encoded libraries (DELs) represent a powerful screening technology for discovering small molecules with high affinity for target proteins[1,2]. Constructed through split-pool synthesis, DELs generate vast combinatorial libraries, often containing billions to trillions of compounds, each uniquely labeled with a DNA barcode. These DNA-tagged compounds are then screened based on their binding affinities to specific target proteins, facilitating high-throughput identification of potential drug candidates. DEL technology has contributed to the discovery of numerous hit compounds in target-based drug discovery, with notable successes including inhibitors for SARS-CoV-2 3CLpro[3], soluble epoxide hydrolase[4], Autotaxin[5], and receptor-interacting protein kinase 1[6].

Despite the high-throughput capabilities and economic advantages of the DEL technique, the number of DEL-derived molecules advancing to clinical trials or reaching the market remains relatively low, partly due to the relatively recent adoption of DEL as a mainstream approach for hit identification, as well as persistent challenges such as uncertainties in processing DEL data, difficulties in optimizing DEL hits, and a general lack of understanding on target druggability with DEL molecules[7]. To overcome these barriers, efforts to integrate artificial intelligence (AI) with DEL screening have emerged, with most focusing on how to select more promising hits from the highly noisy screening data[8]. Notably, DEL molecules are characterized by common features constrained by solution chemistry and the structural requirements for attaching DNA tags, which may result in interaction patterns with target protein pockets. A perspective on the characteristics of protein pockets that bind DEL molecules could provide insights into DEL, potentially enhancing its efficiency and success rates in drug discovery campaigns.

[1]State Key Laboratory of Gene Expression, School of Life Sciences, Westlake University, Hangzhou, Zhejiang, China. [2]Westlake AI Therapeutics Lab, Westlake Laboratory of Life Sciences and Biomedicine, Hangzhou, Zhejiang, China. ✉e-mail: huangjing@westlake.edu.cn

Protein language models (PLMs) have emerged as powerful tools in biological research, with wide applications in protein structure prediction[9], property prediction[10], function annotation[11,12], and protein design and engineering[13]. Despite these advances, large-scale language models specifically designed for binding pockets—the fundamental functional units in drug design—are still limited. Existing methods for pocket representation, such as MASIF[14], primarily rely on learning chemical and geometric features on protein surfaces. In certain machine learning (ML) models for drug–target interaction (DTI) prediction, pocket structures are explicitly encoded[15–18]. For example, Uni-Mol[15] utilizes self-supervised masked atom prediction to learn the representations of pocket structures, while PocketAnchor[16] represents pockets using anchor points sampled in space for downstream pocket detection and binding affinity prediction tasks.

In contrastive learning, ML models are trained to distinguish between similar and dissimilar pairs of data, with the aim often being to learn a generalizable feature representation. Combining such a self-supervised representation learning technique with a pre-trained large PLM, such as ESM-2, enables zero-shot or few-shot learning by leveraging the evolutionary information encoded in the PLMs[19]. This approach has been effectively utilized in DTI prediction by methods such as DrugLAMP[17] and PocketDTA[18]. However, functional classification models for binding pockets remain largely absent.

Key challenges include the lack of comprehensive pocket databases and the inherent structural flexibility of binding pockets, which poses significant difficulties for structure-based models, limiting their effectiveness for functional annotation and classification. Research has also shown that binding pockets for the same ligand (e.g., ATP) can exhibit notable geometric differences[20], and that global structural similarity in proteins does not always correspond to local similarities in pocket structures[21]. These observations underscore the limitations of current pocket representation methods, particularly in distinguishing functionally similar pockets. Recent work has highlighted that refined pocket representations can directly enable the biological discoveries[22]. To address these challenges, a more tailored, function-driven approach to pocket modeling is needed to advance both the understanding of binding pockets and drug discovery.

In this work, to better represent protein pockets and assess their functional similarity, we introduce ErePOC, a pocket representation model based on contrastive learning. In this approach, ESM-2 embeddings are processed through a feedforward neural network (NN) to generate function-aware representations by minimizing a contrastive loss function that increases the similarity between pockets bound to similar ligands while maximizing the dissimilarity between those bound to dissimilar ligands. ErePOC enables the visualization of pocket feature distributions across both experimentally determined and predicted protein spaces, revealing distinct pocket patterns shared by targets binding the same class of ligand. Building upon these capabilities, we focus on exploring the common features of DEL target pockets. By leveraging ErePOC's ability to model these pockets, we screen the human proteome to identify targets containing pockets similar to those successfully engaged by DEL molecules, calculate corresponding enrichment scores, and uncover potential targets suitable for DEL screening. This work not only introduces an approach to understanding and improving DEL-based drug discovery but also enhances target identification and pocket analysis, which are crucial for biological discovery in general.

## Results

This study aims to identify shared characteristics of protein targets suitable for DEL screening, with a particular emphasis on binding pockets as the central unit of analysis. We integrated data from several sources, including BioLiP2[23] and AlphaFill[24], which contain experimental and predicted ligand–protein complex structures, respectively. We also curated two datasets containing complex structures with ligand being either DEL molecules or FDA-approved drugs (FDA-AD). We systematically examined the binding pocket characteristics across the DEL, FDA-AD, and BioLiP2 datasets, focusing on sequence features, physicochemical properties, and binding interactions. The findings are structured as follows: We begin with a detailed analysis of DEL pocket patterns, then introduce the training and validation of the ErePOC model for representing protein pockets. This is followed by an exploration of pocket landscape clustering, where we compare experimentally determined and computationally predicted structures. Finally, we predict human protein classes most likely enriched in DEL screening and evaluate their functional roles and structural similarities at both global and local levels.

### Sequence analysis on pocket datasets

We compared pocket sizes across various structures by analyzing the distribution of amino acid residues within the pockets. These structures are sourced from four categories: (i) the BioLiP2[23] database, from which we selected entries annotated with regular ligands (i.e., biologically relevant small molecules) and their associated pockets; (ii) the AlphaFill dataset[24], which contains computationally predicted ligand–protein complex structures; (iii) a DEL dataset curated from ligands reported to be identified by DEL screening (Supplementary Information Table S1); and (iv) an FDA-AD dataset containing FDA-approved drugs with experimentally determined complex structures (Table S2). They include 326,416, 293,019, 128, and 340 pockets, respectively. For BioLiP2, pockets were defined using experimentally annotated binding residues provided by the web server. Meanwhile, pockets for the AlphaFill, DEL, and FDA-AD datasets were generated by including all amino acid residues within 5 Å of the bound ligand. To assess the impact of this inconsistency, we additionally redefined BioLiP2 pockets using the same distance-based criterion and repeated all analyses under this unified definition. The key findings remained consistent across definitions, suggesting that our conclusions are reasonably robust to differences in pocket definition.

As shown in Fig. 1A, the average number of residues in BioLiP2, AlphaFill, DEL, and FDA-AD pockets was 12.5, 12.5, 28.1, and 16.1, respectively. The larger number of residues surrounding DEL and FDA-AD ligands likely reflects their greater molecular size and chemical complexity, often containing halogen atoms and other bulky functional groups that demand more spatially extended binding environments. The average molecular weights of DEL and FDA-AD ligands in this study are 560.5 and 310.9 Da, respectively (Fig. S2). In contrast, regular ligands and their pockets have co-evolved for an optimized fit tuned to biological needs rather than maximal binding. Synthetic drug molecules are often optimized for potency and selectivity through medicinal chemistry efforts, which typically results in larger and more chemically complex scaffolds compared to regular ligands[25,26]. They often target protein pockets that are more spacious, flexible, and dynamic, accommodating a broader range of interactions.

We analyzed amino acid frequencies in binding pockets across the BioLiP2, DEL, and FDA-AD datasets. To highlight compositional differences, we calculated the enrichment ratio of each amino acid relative to its abundance in the PDB. As shown in Fig. 1B, methionine, tyrosine, tryptophan, and phenylalanine were the four most significantly enriched amino acids in the DEL dataset. These four were also the most enriched in FDA-AD, appearing more than twice as frequently in drug-binding pockets compared to proteins in general. These bulky side chains potentially provide unique pocket geometries for specific molecular binding, as well as anchors for hydrophobic and aromatic interactions. Compared to BioLiP2, hydrophobic amino acids, including methionine and leucine, were significantly enriched in DEL and FDA-AD. In contrast, cysteine showed significantly lower enrichment in drug-binding pockets. We note that our analysis excluded covalent drug molecules, which primarily react with cysteine's thiol group. Overall, our analysis reveals that DEL and FDA-AD pockets

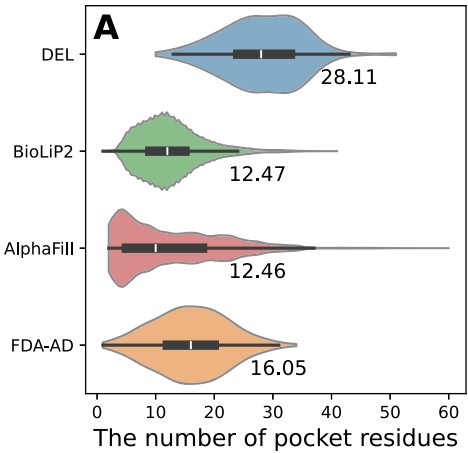

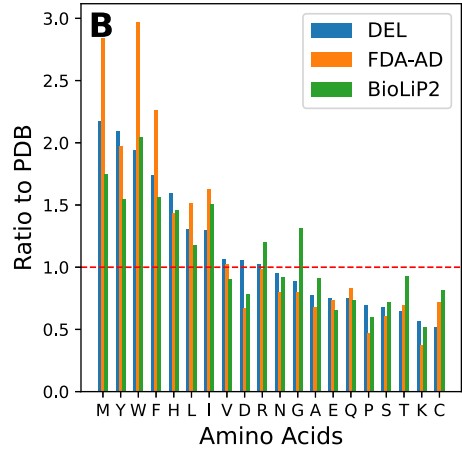

**Fig. 1 | Pocket-size distribution and amino acid frequency analysis. A** Violin plots show the distribution of the number of pocket residues across four datasets: DEL, FDA-AD, BioLiP2, and AlphaFill. The width of each violin represents the kernel density of the distribution. The central line indicates the median, and numerical labels indicate the mean number of pocket residues for each dataset. Sample sizes (*n*) are provided in the Source data file. No statistical hypothesis testing was performed. **B** Bar plots show the relative frequency of the 20 amino acids in the DEL, BioLiP2, AlphaFill, and FDA-AD datasets, normalized by their corresponding frequencies in PDB.

share similar amino acid composition patterns, distinguishing them from pockets that bind regular ligands.

## Analysis of pocket physicochemical properties and their interactions with ligands

We used Fpocket[27] to analyze the biochemical and biophysical properties of pockets across the DEL, FDA-AD, and BioLiP2 datasets. Six Fpocket descriptors were grouped into three clusters to compare pocket size, hydrophobicity, and polarity (Fig. 2A–F). Pocket size was assessed using volume and the number of alpha spheres. The average volumes of pockets were 3301.2 Å³ for DEL, 2534.1 Å³ for FDA-AD, and 2739.5 Å³ for BioLiP2. DEL pockets were approximately 1.3 times larger than FDA-AD pockets and nearly 1.2 times of BioLiP2 pockets. DEL pockets also contained more alpha spheres (164.3) than FDA-AD (118.8), making them 1.4 times larger in this size metric. The average alpha sphere densities were 11.0 Å for DEL, 10.0 Å for FDA-AD, and 10.5 Å for BioLiP2, suggesting that DEL molecules generally bind to more open and exposed pockets.

DEL and FDA-AD pockets are more hydrophobic than BioLiP2 pockets, as indicated by their higher proportions of apolar alpha spheres and larger mean local hydrophobic densities. Interestingly, while DEL and FDA-AD pockets exhibit similar distributions of local hydrophobic densities, their apolar sphere proportions differ, averaging 53.9% for FDA-AD, 50.8% for DEL, and 46.2% for BioLiP2. For polar interactions, the distributions were similar. BioLiP2 had the highest proportion of polar atoms in pockets (38.6%), followed by DEL (37.3%) and FDA-AD (36.0%).

We further analyzed the interactions between pocket residues and ligands using the Arpeggio method[28] (Fig. 2G–I). Hydrophobic interactions accounted for 50.7% in DEL, 42.9% in FDA-AD, and 32.5% in BioLiP2. Hydrogen bond interactions were 3.8% for DEL, 6.7% for FDA-AD, and 9.7% for BioLiP2, while polar interactions constituted 6.0%, 11.7%, and 14.5%, respectively. Ionic interactions were slightly more prevalent in DEL pockets (1.3%) than in FDA-AD pockets (0.7%), while BioLiP2 pockets exhibited a higher fraction of ionic interactions (3.9%). This pattern may reflect the early, pre-optimization status of DEL compounds compared to approved drugs.

Overall, these results indicate that DEL hits preferentially bind to larger, more hydrophobic pockets. The expansive contact area in these pockets enhances binding through shape complementarity, thereby favoring hydrophobic interactions. These characteristics suggest potential avenues for optimization toward drug-like molecules, for example, by balancing polar interactions to improve binding specificity. Statistical analyses were performed by computing Cliff's δ effect sizes for key pocket and pocket–ligand interaction features (Fig. S3). These analyses confirmed that DEL pockets were significantly larger than FDA-AD targets (volume $\delta = 0.405$; number of alpha spheres $\delta = 0.409$; alpha sphere density $\delta = 0.395$; $p < 3.6e-11$) and BioLiP2 pockets (volume $\delta = 0.302$; number of alpha spheres $\delta = 0.321$; alpha sphere density $\delta = 0.201$; $p < 3.4e-09$), and exhibited a balanced polar-apolar composition, with physicochemical properties intermediate between FDA-AD and BioLiP2 (Fig. S3A). Analysis of pocket–ligand interactions confirmed that DEL pockets were dominated by hydrophobic contacts ($\delta = 0.122$ vs FDA-AD, $\delta = 0.378$ vs BioLiP2), whereas hydrogen bonds ($\delta = -0.150$ vs FDA-AD, $\delta = -0.392$ vs BioLiP2) and polar interactions ($\delta = -0.207$ vs FDA-AD, $\delta = -0.459$ vs BioLiP2) were reduced (Fig. S3B). These characteristics indicate that DEL binding is primarily driven by hydrophobic effects, stabilized by a minimal yet functionally critical set of polar anchors.

Principal component analysis (PCA) of features further confirmed these patterns, showing that DEL pockets occupy a distinct region in PCA space (Fig. S4). PC1 primarily reflected chemical composition, including apolar/polar atom ratios and interaction types, while PC2 was dominated by structural size descriptors, together accounting for approximately 75% of the variance. The pocket analysis was also consistent with molecular property differences obtained using Molecular Operating Environment (MOE)[29] (Fig. S2). DEL molecules exhibited lower water solubility (LogS = −6.49) compared to approved drugs (−3.05) and higher hydrophobicity (cLogP = 3.42 vs. 1.44). While DEL pockets shared the overall druggability characteristics of FDA-AD targets, they displayed a distinct physicochemical bias, favoring larger, more hydrophobic pockets with a different polar–apolar balance. We note that no single feature or simple combination could distinguish DEL from FDA-AD or general protein pockets, possibly due to the broad variability of pocket structures. This observation highlights the need for developing a more informative representation of pockets.

## ErePOC: an Enhanced representation of POCkets by contrastive learning

We developed ErePOC, a contrastive learning-based model trained on 326,416 pocket–ligand pairs from the BioLiP2 dataset, to generate latent space representations of binding pockets (Fig. 3). For each pair, pocket residues were embedded using 1280-dimensional feature vectors from ESM-2, and pocket-level features were obtained by average

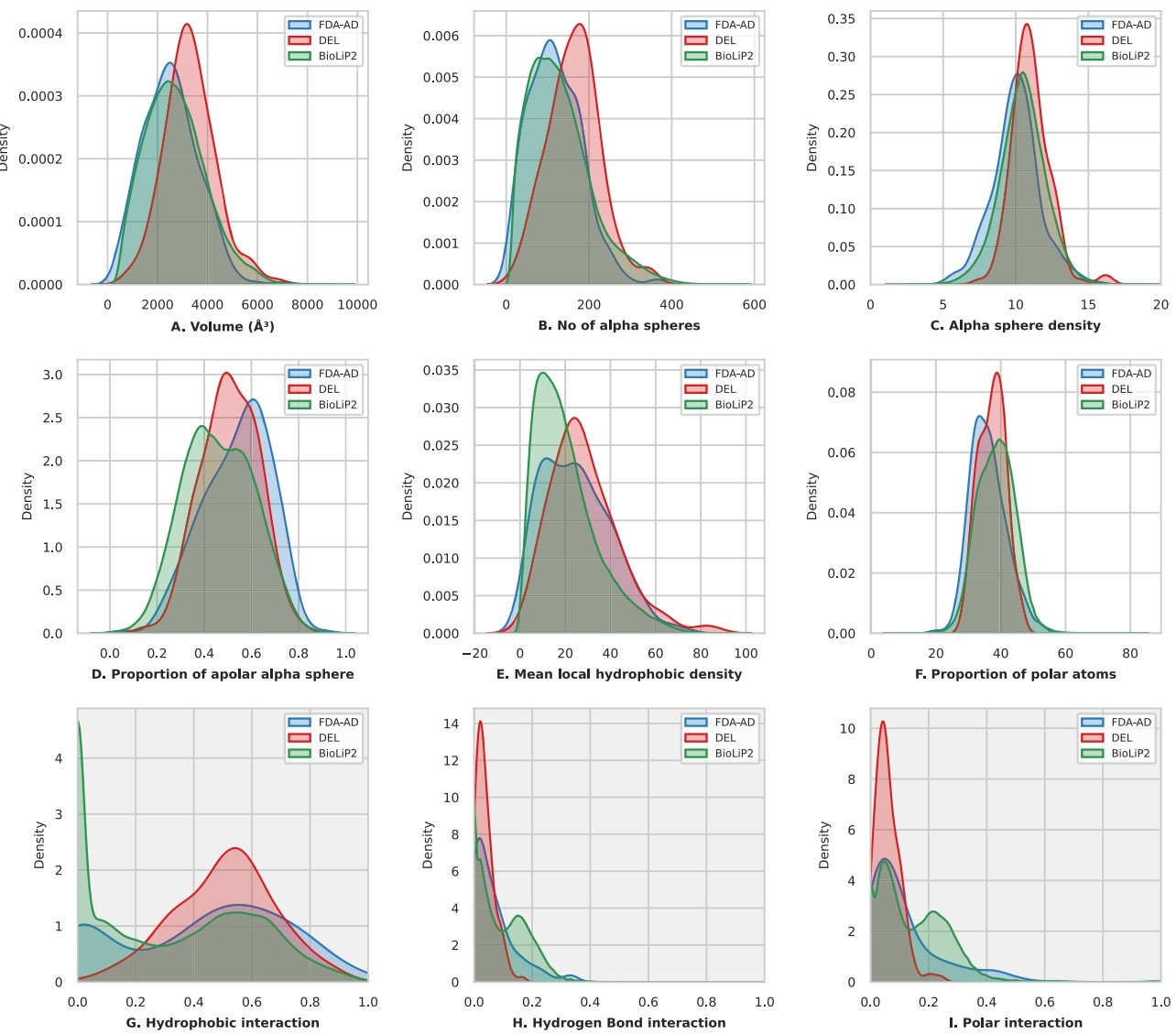

**Fig. 2 | Physicochemical properties of pockets and ligand–pocket interaction analysis for DEL, FDA-AD, and BioLiP2 Datasets. A–F** The physicochemical properties of pockets computed using Fpocket, including volume, number of alpha spheres, the density of alpha spheres, the proportion of apolar alpha spheres, mean local hydrophobic density, and the proportion of polar atoms. **G–I** The ligand–pocket interaction characteristics, focusing on hydrophobic interactions, hydrogen bonds, and polar interactions. Sample sizes (*n*) are indicated in the Source data.

pooling of residues forming the binding pocket. This approach ensures that the spatial profile captures structural information intrinsic to the pocket. To quantify pocket similarity, each 1280-dimensional pocket embedding was projected into a 256-dimensional latent space through a two-layer perceptron. Pocket similarity was computed as the cosine similarity between these 256-dimensional pocket representations, while ligand similarity was evaluated using the cosine similarity of their 2048-dimensional molecular fingerprints. Unlike conventional DTI models, ErePOC employs Kullback-Leibler (KL) divergence as the loss function to align ligand distributions with the pocket landscape, encouraging pockets that bind chemically similar ligands to be positioned closer in the latent space.

Ultimately, ErePOC learns a compact 256-dimensional representation for each pocket, effectively capturing both fine-grained similarities and key distinctions among binding sites. By aligning pockets according to the chemical similarity of their ligands through contrastive learning, this approach results in an enhanced, function-aware representation of binding pocket features. The resulting latent space provides deeper insights into pocket–ligand interactions, enhancing the classification and analysis of binding pockets.

We evaluated our model using zero-shot learning tasks to compare the performance of representations derived from ESM-2 embeddings and those learned by ErePOC. We considered a classification task involving seven types of pockets, each corresponding to a unique ligand type—ADP (9531 pockets), COA (1900), FAD (6367), HEM (13,312), NAD (5354), NADP (3997), and SAM (1228)—totaling approximately 43,000 binding pockets curated from BioLiP2. Figure 4A, B shows the clustering of these seven binding pocket types using t-SNE[30] based on ESM-2 and ErePOC representations, respectively. The results clearly demonstrate that the contrastive learning framework generates well-separated clusters for different ligand/pocket types, effectively capturing both functional and ligand-specific features of the binding pockets. In contrast, the ESM-2-based model, which lacks functional annotations specific to pockets, shows limited separation between pocket types. This comparison highlights the superior performance of contrastive learning in producing more refined and informative representations for functional pocket classification.

To assess the robustness of our model, we conducted an ablation study in which two types of binding pockets were completely excluded from the BioLiP2 training dataset prior to contrastive learning. We then

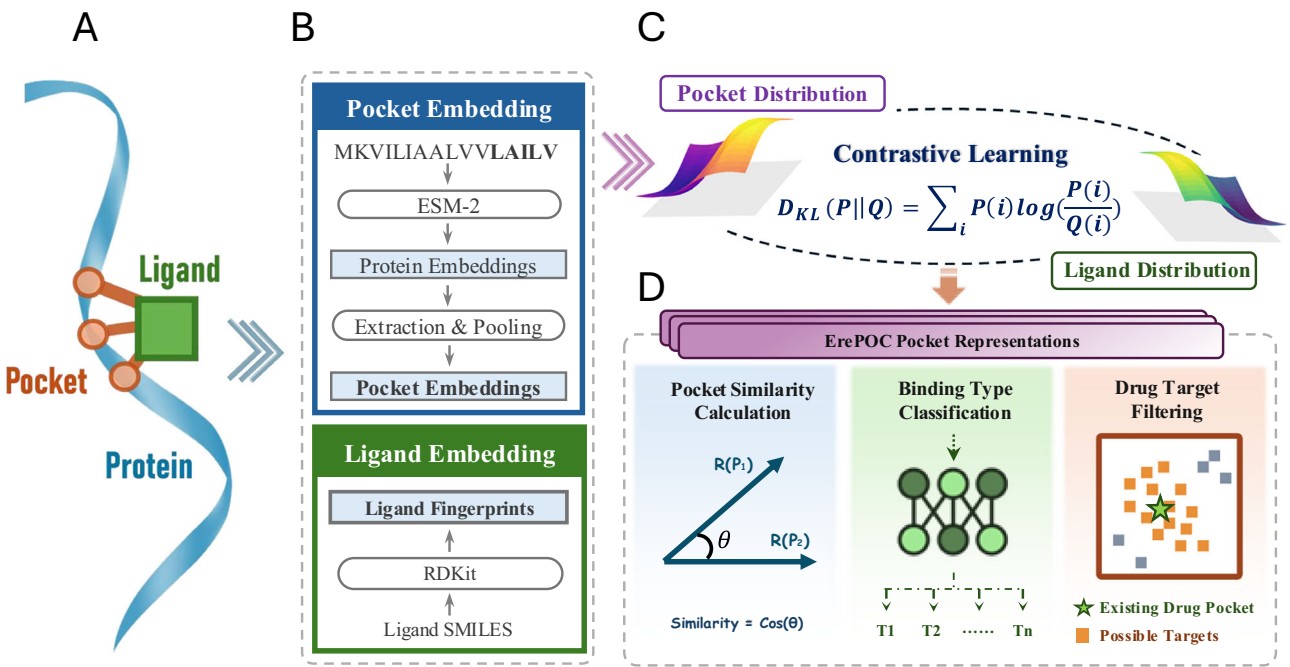

**Fig. 3 | The ErePOC pipeline. A** During training, pocket residues are extracted by identifying the residues surrounding the ligand within a specific distance cutoff in the protein-ligand complex structure. **B** The entire protein sequence is embedded using ESM-2, producing a 1280-dimensional vector for each residue. The pocket residues are then selected, and their average vector is used as the pocket's ESM-2 embedding feature. The ligand's 2048-dimensional fingerprint is computed using RDKit. **C** $Q(i)$ represents the distribution of ligand similarity, while $P(i)$ denotes the distribution of pocket similarity, derived from vectors learned through contrastive learning. The KL divergence is used as the loss function to align $P(i)$ with $Q(i)$. **D** A 256-dimensional vector from the final latent space of the contrastive learning model is used as the ErePOC representation. The ErePOC representation model enables three important downstream applications: functional binding evaluation through cosine similarity, binding type categorization, and drug–target identification.

evaluated the model's performance in classifying these excluded pocket types. Figure S5 presents t-SNE visualizations for various exclusion scenarios: ADP and FAD, HEM and ADP, ADP and NAD, and HEM and SAM pockets. The results show that strong classification performance was maintained even for pockets excluded from training. The contrastive learning framework effectively distinguishes the removed ligand types, highlighting its ability to generalize and accurately classify pockets based on their functional and ligand-binding features. This robustness suggests that the model leverages annotated ligand information during training to make reliable predictions, even when specific ligand types are absent from the training set. This underscores the effectiveness of ErePOC in capturing and generalizing key binding pocket characteristics.

The approach was further validated by analyzing ATP-, FAD-, and HEM-binding pockets from both the BioLiP2 dataset (experimentally determined structures) and the AlphaFill dataset (which imputes ligands into AF2-predicted structures). t-SNE clustering based on ESM-2 features (Fig. S6A) showed limited separation between pocket types. In contrast, the ErePOC representation revealed substantial overlap among pockets binding the same ligands across both datasets, demonstrating ErePOC's ability to capture structural similarities between experimental and predicted protein-ligand complexes.

Additionally, 500 randomly selected pockets from the BioLiP2 dataset were used to compute the correlation coefficient (shown in Fig. S7). Pearson correlation analysis demonstrated a robust correlation of 0.96 between ligand Tanimoto similarity and pocket cosine similarity derived from the ErePOC vector, highlighting ErePOC's aptitude for capturing meaningful pocket–ligand interactions. Furthermore, five pockets were randomly chosen from each of the seven ligand-binding types to compute pairwise cosine similarities. The heatmap in Fig. S8 compares the similarity results using the ESM-2 embeddings, ErePOC vectors, and t-SNE 2D projections following ErePOC transformation. The cosine similarity computed from ErePOC

representations effectively distinguished between various pocket types, whereas ESM-2 demonstrated limited discriminatory power. Moreover, cosine similarity based on t-SNE-reduced representations also performed poorly in assessing pocket relationships. In summary, ErePOC proves highly proficient in identifying pockets that bind ligands with analogous structural features.

We designed another downstream classification task involving the prediction of the seven ligand-binding pocket types using few-shot learning with ESM-2 and ErePOC representations. For independent testing, 10% of the targets were held out, ensuring a comprehensive evaluation of the model's performance. Given the relatively straightforward nature of this classification task, all models achieved strong results. In this few-shot learning setup, we tested four models: ErePOC-NN, ErePOC-SVM, ESM-2-NN, and ESM-2-SVM. The ErePOC-NN and ErePOC-SVM models used pocket representations derived from contrastive learning as input features, paired with either an NN or a support vector machine (SVM) classifier, respectively. Similarly, ESM-2-NN and ESM-2-SVM relied on direct embeddings from ESM-2, utilizing NN and SVM classifiers.

Figure S9 compares the performance of these models on the testing dataset. ESM-2-NN achieved the highest overall accuracy (0.989), followed closely by ErePOC-NN (0.986) in classifying the seven ligand-bound pocket types. We note that the MaSIF-ligand model, trained using the MaSIF representation, achieved an accuracy of 0.74 for the same task, although the result was obtained on a different test set[14]. Interestingly, when assessing the performance of the SVM models with an RBF kernel, ESM-2-SVM's accuracy dropped significantly to 0.811, while ErePOC-SVM maintained a high accuracy of 0.985. This notable difference underscores the superiority of contrastive learning in generating robust representations for functional pocket classification. It also highlights ErePOC's capability to generalize to diverse or previously unseen pockets, whereas ESM-2's pretrained features appear less effective for this specific task without

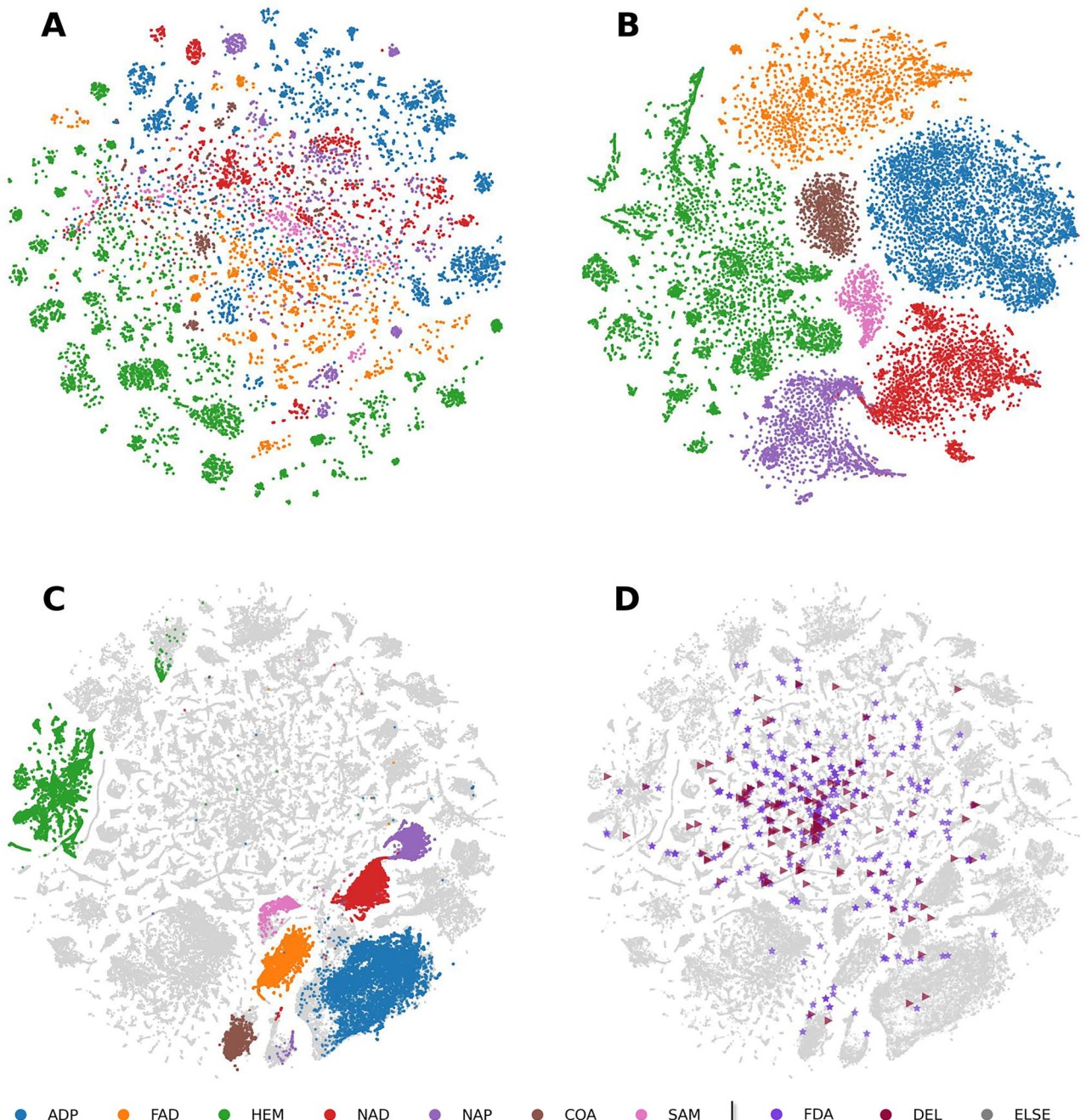

**Fig. 4 | t-SNE visualization of ErePOC and ESM-2 representations for the BioLiP2 dataset.** Visualization of the 7 types of ligand-binding pocket landscapes using ESM-2 (**A**) and ErePOC (**B**) representations, respectively. **C** Pocket landscape using ErePOC representations. **D** Comparison of pocket landscapes for the FDA-AD and DEL datasets against BioLiP2 using ErePOC representations. The pockets include ADP ($n = 9513$), FAD ($n = 6367$), HEM ($n = 13,312$), NAD ($n = 5354$), NAP ($n = 3997$), COA ($n = 1900$), and SAM ($n = 1228$); FDA-AD ($n = 340$) and DEL ($n = 128$).

further fine-tuning. We note that for ErePOC model training and downstream analyses, the original BioLiP2 binding residue annotations were used to preserve experimentally validated contacts. Training and analyses using the alternative 5 Å pocket definition yielded highly similar results, as demonstrated in Figs. S8–S10.

### Clustering and characterization of DEL pockets in experimental and predicted protein landscapes

The primary goal of this study is to explore the distribution of drug-related and lead-binding pockets across the entire protein space. Using the ErePOC representation, we projected the FDA-AD and DEL datasets onto the comprehensive pocket landscape generated using the

BioLiP2 dataset (Fig. 4C, D) and the AlphaFill dataset (Fig. S11). t-SNE visualization illustrates that experimentally resolved metabolite pockets are organized into distinct localized regions (Fig. 4C), demonstrating ErePOC's ability to differentiate functional pockets. Furthermore, pockets bound to approved drug molecules (FDA-AD, shown in purple in Fig. 4D) are widely distributed throughout the protein space, rather than being confined to specific clusters, highlighting their versatility. Figure 4D also highlights the similar distribution patterns of DEL pockets (dark gray) and FDA-AD pockets, which scatter across the latent space. This spatial concordance aligns with previous Fpocket analyses, which emphasized similarities in their physicochemical properties. Together, these results suggest that DEL

screening can access the majority of known druggable targets, while exhibiting a measurable preference for pockets with features conducive to DEL hit generation, such as larger volume and higher hydrophobicity.

The ErePOC representation partitions the BioLiP2 pocket space into distinct patterns, offering critical insights into the global pocket landscape. For example, pockets bound to native ligands such as SAM or HEM are notably absent from the DEL and FDA-AD chemical spaces, suggesting that these tightly bound, cofactor-associated pockets may be less suitable for conventional DEL screening. To further explore the "DEL-able" regions of the pocket landscape, we identified the five nearest neighbors for each DEL target in the BioLiP2 dataset based on cosine similarity (Fig. S12). A total of 361 pockets in BioLiP2, referred to as DEL neighbors, exhibited cosine similarity scores greater than 0.8. The physicochemical properties of these DEL neighbors, as computed using Fpocket, are summarized in Fig. S13. The average pocket volume for the DEL neighbors changed from 1612.84 to 2038.69 $\text{Å}^3$, corresponding to an approximate 26.4% increase relative to the average volume of natural pockets in the BioLiP2 dataset. The average number of alpha spheres for DEL neighbors changed from 69.35 to 92.35, reflecting a 33.2% increase and indicating higher structural complexity. Furthermore, the mean local hydrophobic density for DEL neighbors increased from 14.98 to 21.71, representing a 44.9% enhancement, emphasizing their pronounced hydrophobic nature.

In terms of interaction profiles, hydrophobic interactions in DEL neighbors changed from 30.1 to 42.0, reflecting a 36.7% increase compared to BioLiP2 pockets. Hydrogen bond interactions, however, decreased from 9.9 to 7.0. Polar interactions in DEL neighbors increased from 9.8 to 14.8, showing a 51.0% rise in polarity relative to the BioLiP2 pockets. In total, these DEL neighbors exhibit larger volumes and greater spatial expansiveness than typical natural pockets in BioLiP2, with physicochemical properties that closely resemble those of both DEL and FDA-AD pockets. The similarities of DEL neighbors make them promising candidates for DEL screening. Furthermore, the ligand–pocket interaction profiles of DEL neighbors, DEL pockets, and FDA-AD pockets were found to be comparable, as illustrated in Fig. S13G-I, reinforcing the similarity in their functional characteristics.

### Identification and characterization of human protein pockets for DEL screening

We screened AlphaFold2-predicted human proteins (23,391) to predict binding pockets most likely suitable for DEL screening[31]. First, Fpocket was used to identify pockets in these human proteins. For each pocket, the average pLDDT of its constituent residues was calculated as the pocket pLDDT score. Pockets with a volume smaller than 800 $\text{Å}^3$ or a pLDDT score below 0.7 were excluded, yielding a total of 182,424 human protein pockets. The 800 $\text{Å}^3$ threshold was selected based on prior studies suggesting 500 $\text{Å}^3$ as a minimal druggable pocket volume[32], together with our observation that DEL-binding pockets are substantially larger. These pockets were then encoded using ErePOC embeddings, and their cosine similarity to 128 known DEL pockets was computed, from which the highest similarity was assigned to each pocket. Ultimately, 4774 pockets with a cosine similarity greater than 0.8 were identified. After removing duplicates based on UniProt ID, 2739 unique human proteins were predicted to contain DEL-compatible pockets. The overall prediction workflow is illustrated in Fig. 5A.

Among the predicted human proteins, 17.9% were classified as transferases, 11.6% as hydrolases, and 9.4% each for DNA-binding proteins and oxidoreductases. In comparison, the 128 DEL target proteins were predominantly transferases (27.1%), followed by hydrolases (17.4%) and receptors (9.7%). The FDA-AD dataset, which encompasses 31 protein families, showed a similarly broad distribution, with transferases (20.8%), hydrolases (18.1%), oxidoreductases

(14.8%), DNA-binding proteins (7.3%), and receptors (6.9%) as the most frequent classes. Enrichment scores for each protein class were calculated using a hypergeometric test, and Fig. 5C depicts the distribution of enrichment scores for proteins with $p$ values < 0.05. Notably, several classes, including oxidoreductases, multifunctional enzymes, transferases, chromatin regulators, lyases, and isomerases, exhibited enrichment scores ranging from 1.36 to 6.24 in both the DEL target set and the predicted human protein dataset. Interestingly, transferases, hydrolases, and oxidoreductases are consistently enriched across all three datasets (DEL, FDA-AD, and predicted DEL-compatible pockets), underscoring their central roles as preferred families for DEL screening.

Among known DEL targets, three multifunctional enzymes—AASS (aminoadipate-semialdehyde synthase, PDB: 5L78), PSMA (proteasome subunit alpha, PDB: 1Z8L), and sEH (soluble epoxide hydrolase, PDB: 1ZD4)—resulted in the strongest enrichment. These enzymes catalyze two or more distinct biochemical reactions, which explains their dual classification in our analysis (e.g., sEH is counted as both a hydrolase and an oxidoreductase). The significant enrichment of multifunctional enzymes ($p < 0.05$) suggests that DEL molecules may preferentially target proteins with complex structural architectures and flexible pockets that support multiple catalytic activities.

Figure S14 presents a t-SNE visualization of the distribution of DEL pockets within human proteins, alongside human protein pockets with cosine similarity >0.8. Consistent with findings from the BioLiP2 and AlphaFill datasets, DEL pockets exhibit a broad and diverse distribution. Notably, human protein pockets closely resembling DEL pockets cluster into three distinct groups. However, a substantial number of DEL pockets lack highly similar counterparts among human proteins. This disparity may arise from the limitations of AlphaFold2 in predicting accurate protein structures or the potential inaccuracies of Fpocket in identifying binding pockets, both of which could impact the detection of DEL-like pockets across the human proteome.

### Global and local structure comparison for human DEL targets

Figure 6 presents case studies comparing global and local pocket structures for protein classes enriched in both predicted and known DEL targets. Using ErePOC embeddings, the cosine similarity scores of binding pockets within each class were calculated. Pockets with high cosine similarity and UniProt IDs assigned to the same protein class were grouped for comparative analysis of their global and local structural features. Global structural similarity was assessed using TM-align[33], which computes the TM score measuring the 3D similarity of overall protein structures, while pocket-level similarity was evaluated using PPS-align[34], which reports the pocket similarity (PS) score measuring the 3D similarity of pockets. Both scores range from 0 to 1, with higher values indicating a more similar topology. Specifically, a PS score greater than 0.46 indicates that the pocket has a similar structure[34]. Illustrated in Fig. 6A–D are four representative cases, each from the oxidoreductase, multifunctional enzyme, transferase, and hydrolase classes, that exhibit high similarity in latent representation space despite dissimilarity in both global protein and local pocket structures. These cases indicate that contrastive learning may capture potential functional or physicochemical relationships between binding pockets that are not fully explained by global protein fold or local geometric similarity. Comparable observations have been reported in earlier studies, where pockets binding the same ligand (e.g., ATP) exhibit considerable geometric diversity[20], and functional associations are detectable across distinct structural folds[35]. Although further experimental evidence will be required to substantiate such relationships in our predictions, these findings suggest that embedding-based similarity could provide information complementary to conventional structure alignment methods and may offer hypotheses for future exploration.

In contrast to the above examples, we also identified instances with consistently high similarity in the latent pocket representation

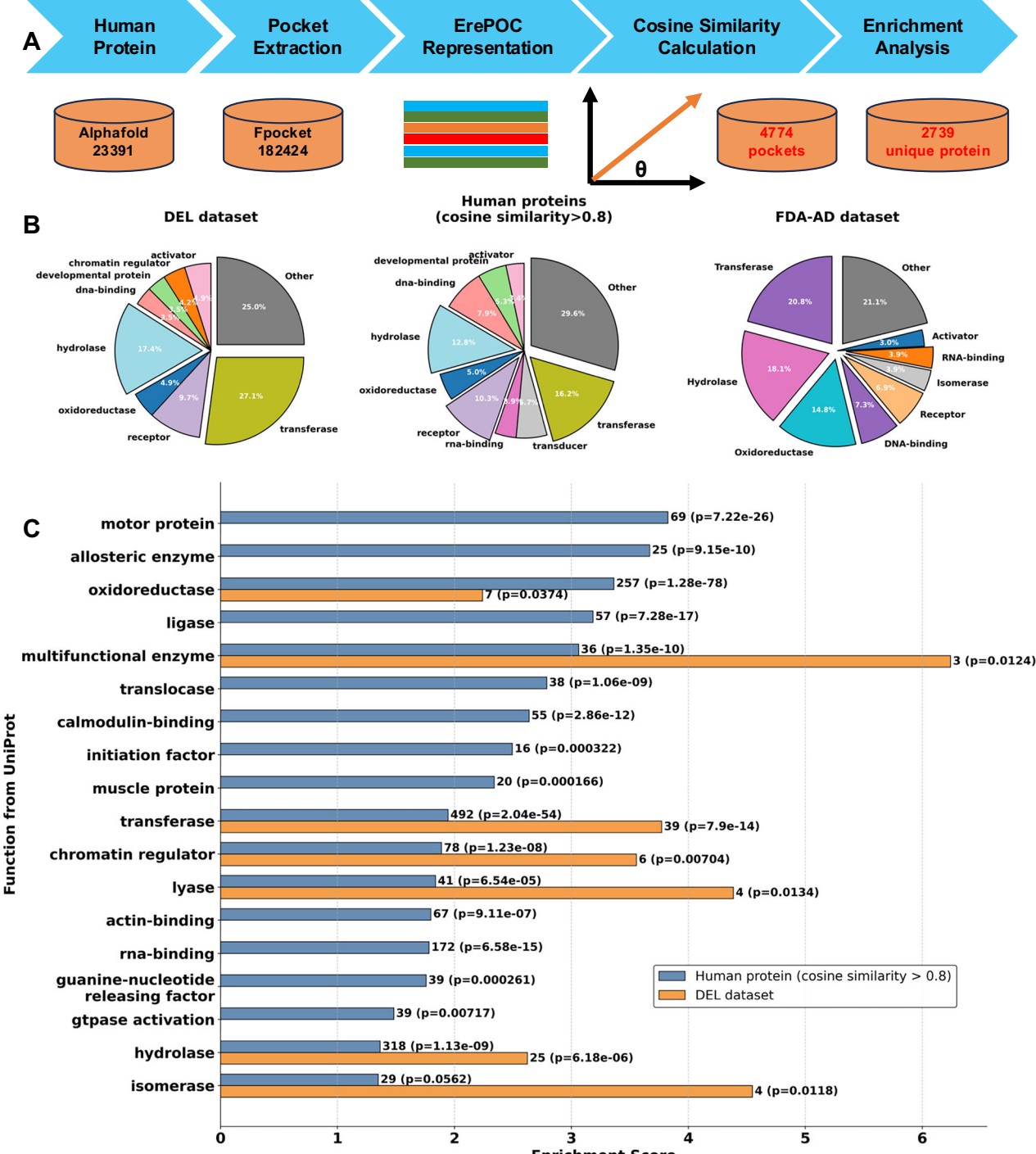

**Fig. 5 | Predicting human protein targets suitable for DEL screening.**
**A** Screening pipeline: a total of 23,391 human proteins predicted by Alphafold were analyzed. Fpocket identified 182,424 pockets, which were represented using Ere-POC embeddings. The cosine similarity between each DEL pocket and the human pockets was calculated, with the highest similarity score used as the final score. Human pockets with a cosine similarity >0.8 to DEL pockets were considered suitable for DEL screening. Enrichment scores for each protein were determined using a hypergeometric test. **B** Proportion of human proteins predicted to have pockets suitable for DEL screening, compared with DEL and FDA-AD targets. **C** Distribution

of enrichment scores for the predicted human proteins with *p* value < 0.05. Enrichment scores were calculated as the ratio of the proportion of proteins with a specific function in the target set (2739 proteins with cosine similarity >0.8) relative to the proportion in the background set (all 23,391 human proteins). Statistical significance was assessed using a one-sided hypergeometric test for over-representation. Bars represent enrichment scores; numerical labels indicate protein counts for each functional category, with exact *p* values shown in parentheses. No adjustments for multiple comparisons were made; *p* values are uncorrected.

space and in global and local structures. Two such examples—one from the lyase class and another from the chromatin regulator class—are illustrated in Fig. 6E, F, exhibiting strong structural similarity at both levels, with TM scores exceeding 0.9 and PS scores above 0.5. We note

that some divergence in pocket residues may arise due to the sensitivity of pocket definition, yet the latent space representation appears robust to such variation. Figure 6G illustrates a case involving a known DEL target (FKBP) and the predicted protein target P26885 (FKBP2),

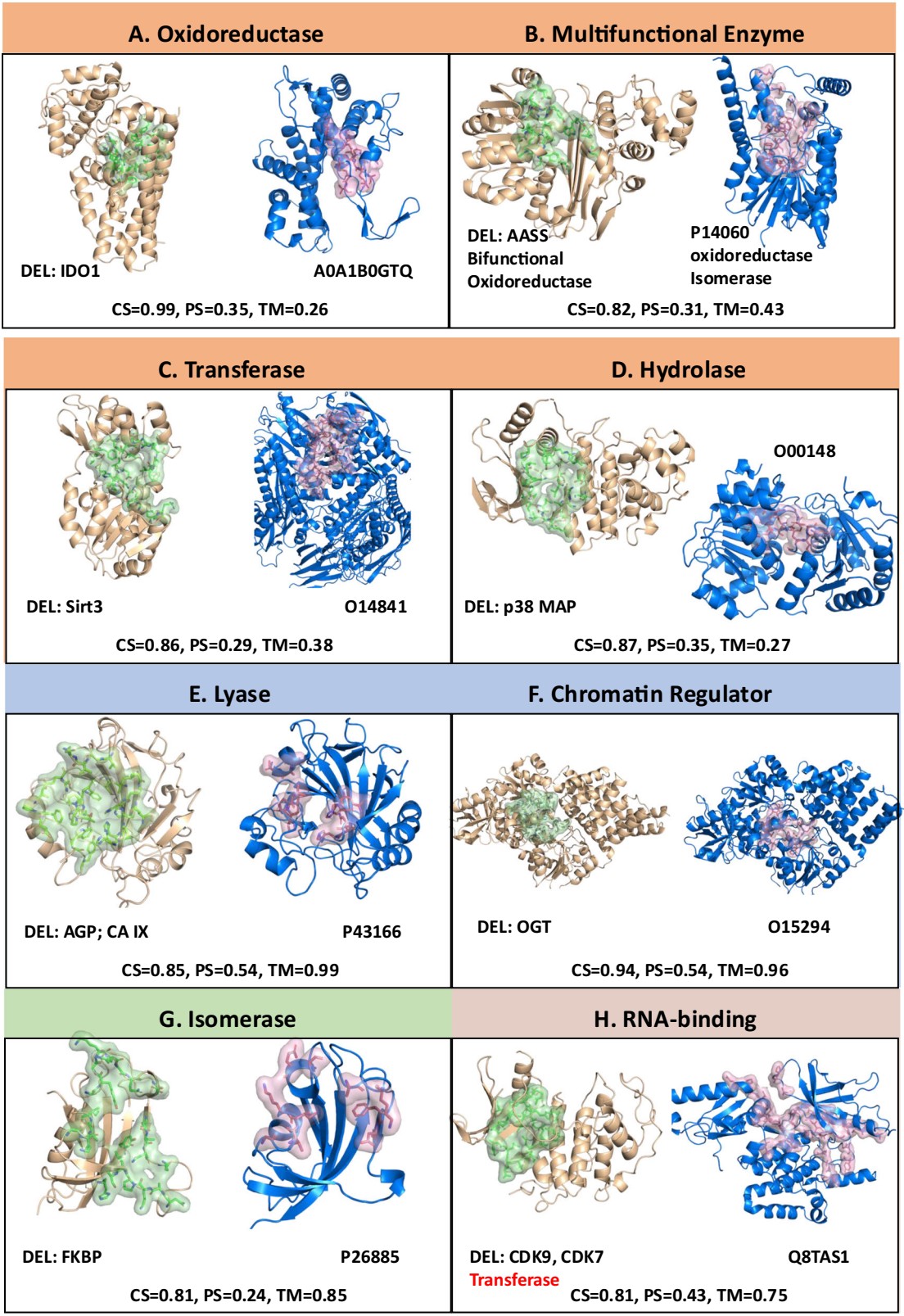

**Fig. 6 | Comparative analysis of global structures and local pocket structures across the enriched DEL targets and human proteins. A–H** Comparisons for each protein class. "CS" represents cosine similarity calculated using ErePOC embeddings, "TM" indicates the TM score obtained through TM-align for whole protein 3D structural comparison, and "PS" denotes pocket structure similarity assessed via PPS-align. All pockets are visualized as a cartoon and surface representation by Pymol. DEL pockets comprise residues within 5 Å of bound ligands, while human protein pockets were predicted using Fpocket.

which shares a TM score of 0.85, indicating strong global structural similarity. However, their pocket similarity score is only 0.24, likely due to the flexible and extended nature of the binding pocket, which severely limits the effectiveness of rigid-body 3D alignment of local structures. Despite this, ErePOC identified a high cosine similarity of 0.81 in the pocket latent space, reasonably suggesting that FKBP2 should also be a target accessible to DEL molecules.

Our analysis is not restricted to the functional classes annotated in UniProt. For example, ErePOC identified potential ligand-binding similarity between the RNA-binding protein NOP56 (UniProt: Q8TAS1) and the SAM-dependent methyltransferase TrmD (PDB: 1UA2), despite their distinct canonical biological roles. A moderate TM score (0.75) suggests a shared Rossmann-like fold, while a moderate PS score (0.43) may arise from conserved cofactor-binding architectures (e.g., SAM/RNA interaction interfaces). This observation implies that DEL-derived chemotypes targeting TrmD's catalytic pocket could potentially engage other RNA-modifying enzymes with analogous structural features.

These findings highlight that while global and pocket-level structural similarities can indicate spatial configurations, they do not fully characterize binding pocket functionality. The inherent flexibility of binding pockets and the conformational selection or induced fit that occurs during ligand binding further complicates their structural representation. ErePOC's robustness stems from its integration of ESM-2 embeddings, which incorporate evolutionary information at the amino acid level to generate high-dimensional representations of physicochemical properties. Moreover, contrastive learning enhances the specific functional characterization, offering resilience to structural distortions induced by ligand binding. These underscore ErePOC's utility in providing a comprehensive and functionally relevant representation of binding pockets.

As a further validation, we designed a large-scale in silico DEL screening experiment against 14 selected human protein targets to compare their capability to bind DEL-like compounds (Fig. S16). Six of these targets, each drawn from a different DEL-enriched functional family identified in our earlier analysis, contained a pocket exhibiting high ErePOC cosine similarity (>0.8) to known DEL pockets. These include a chromatin regulator (O15294), a hydrolase (P03951), an isomerase (P26885), a lyase (P43166), a multifunctional enzyme (P14060), and an RNA-binding protein (Q8TAS1). As a comparison group, six additional targets were selected from DEL-neutral protein families, which did not show enrichment but nevertheless contained pockets with high ErePOC similarity (>0.8). These include a signal transduction inhibitor (O14508), a mitogen (Q9H706), an elongation factor (P43897), an actin capping protein (P47756), a hypotensive agent (P68871), and a cyclin (Q5T5M9). In addition, two closely related proteins, MAT2A (P31153) and MAT2B (Q9NZL9), were included as a family-level case study. The catalytic MAT2A contains a small, polar pocket with a maximum ErePOC similarity of only 0.66 to known DEL pockets. In contrast, the regulatory MAT2B, though non-catalytic, harbors a larger and more hydrophobic pocket with a much higher similarity (0.93), providing an illustrative example of differential DEL compatibility within a single protein family.

Docking calculations were performed using a published virtual DEL compound library (approximately 2.8 million molecules) sampled from 15 sublibraries of the three-cycle HitGen OpenDEL library[36]. No DNA tags were added, so they represent off-bead synthesized molecules. Motivated by previous studies showing that more favorable (i.e., more negative) docking scores correlate with higher experimental hit rates, and that larger libraries improve discovery efficiency[37,38], we evaluated whether ErePOC-identified pockets exhibit similarly favorable trends for DEL compounds. Standardized docking $Z$-score distributions, calculated using the global mean and standard deviation of docking scores across all compounds, are shown in Fig. 7A. Targets from DEL-enriched families exhibited stronger predicted binding, with

more negative mean $Z$-scores (−2.18 vs −1.07) and docking scores (−7.45 vs −6.15 kcal mol⁻¹) compared to those from DEL-neutral families. The difference was also statistically significant when examining the top 1% scoring compounds per target: DEL-enriched targets yielded mean top 1% docking scores ranging from −8.93 to −11.96 kcal mol⁻¹ ($Z = −1.54$ to $−3.73$), whereas DEL-neutral targets ranged from −5.49 to −9.73 kcal mol⁻¹ ($Z = +0.95$ to $−2.12$) (Fig. 7B). Monte Carlo resampling over 1000 iterations (Fig. 7C) confirmed the robustness of these trends across relevant cutoffs (−10 to −8 kcal mol⁻¹), supporting a significant difference in the predicted ability to bind DEL molecules as measure by docking scores. These differences were also evident in the distribution of docking scores (Fig. 7D and Supplementary Fig. S17).

## Discussion

In this study, we investigated the binding patterns of 128 targets that have been successful in DEL screening, focusing on the local structures and properties of their binding pockets. Our analyses included sequence-based evaluations and assessments of physicochemical properties, such as pocket characteristics and ligand–pocket interactions. The results revealed that DEL binding pockets possess structural and chemical features compared to regular ligand-binding pockets. Specifically, methionine and leucine were significantly enriched in DEL binding pockets, likely due to their flexibility, which allows the pocket to adapt its conformation to accommodate diverse ligand shapes. Furthermore, DEL pockets were found to have larger volumes and predominantly hydrophobic environments compared to regular ligand-binding pockets in the BioLiP2 dataset. These insights motivate the key innovation of this study: developing ErePOC to learn high-dimensional pocket representations beyond descriptive features, and applying it to computationally expand the landscape of DEL-suitable targets across the human proteome.

It is well established that structure determines function. However, when focusing on binding pockets—the minimal functional units relevant to drug design—their inherent flexibility presents a significant challenge. Two pockets that bind the same ligand can often display low 3D structural similarity, as reflected by PS scores below 0.4. This underscores the difficulty of assessing pocket similarity based solely on geometries. Furthermore, the functional similarity between pockets is closely tied to local physicochemical and thermodynamic properties, which are inherently complex and challenging to quantify.

As a step toward addressing this issue, we developed ErePOC, a pocket representation model that leverages contrastive learning with ESM-2 embeddings to evaluate pocket similarity. Using t-SNE visualizations of the BioLiP2 and AlphaFill datasets, we showed that ErePOC can reliably classify and distinguish pockets that bind the same ligand type. This capability was enabled by a KL divergence-based contrastive loss that aligns pocket similarity with ligand similarity, allowing functional classification of pockets even in the absence of explicit annotations. For each input protein pocket, ErePOC generates a compressed feature vector that serves as the latent representation that integrates local physicochemical properties and evolutionary constraints. Validation results demonstrated that ErePOC effectively overcomes the limitations of traditional 3D pocket structure alignment. These findings highlight ErePOC's potential as a powerful tool for pocket characterization and its applicability in advancing AI-driven drug design.

Using the cosine similarity calculated by ErePOC, we identified neighboring pockets of DEL pockets from the BioLiP2 dataset and analyzed their physicochemical properties and interaction patterns. The results demonstrated that these neighboring pockets shared highly similar property distributions with DEL pockets, further validating ErePOC's capability to effectively represent pocket features. Extending this analysis to 23,391 AlphaFold2-predicted human protein structures, we identified over 2700 candidates with pockets closely

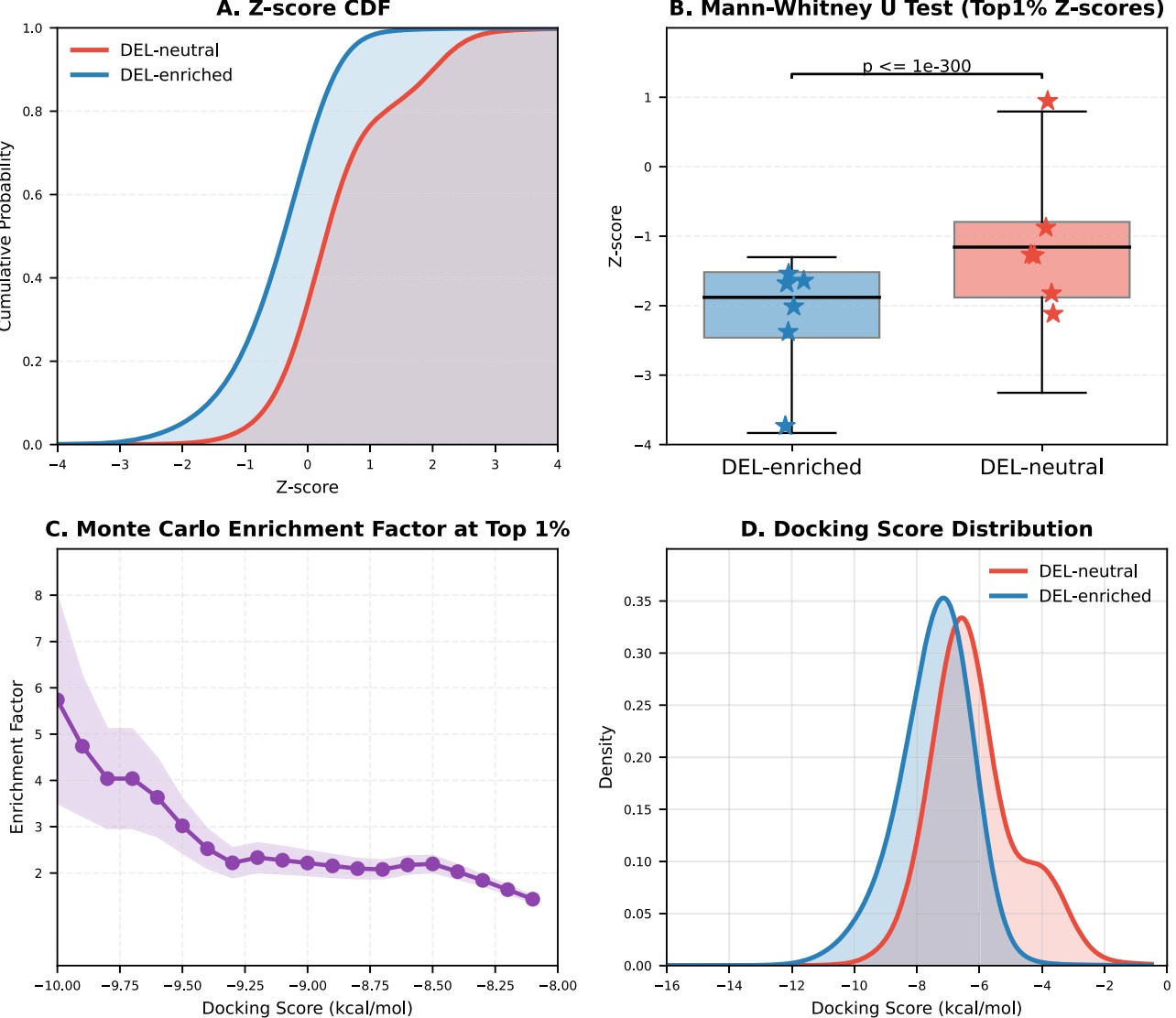

**Fig. 7 | Docking score comparison and enrichment analysis of DEL-enriched and DEL-neutral targets. A** Smoothed cumulative distribution functions (CDFs) of standardized docking $Z$-scores, illustrating a shift toward more favorable binding scores for DEL-enriched targets. **B** Mann–Whitney $U$ test comparing top 1% docking $Z$-scores between 6 DEL-enriched and 6 DEL-neutral groups. Boxes represent the 25th to 75th percentiles, with the median indicated by a horizontal line; whiskers extend to 1.5 × the interquartile range. Stars represent the mean $Z$-score of the top 1% compounds for each target. The analysis includes $n = 186{,}323$ and 193,266 compounds in the DEL-enriched and DEL-neutral top 1% pools, respectively. Statistical significance was assessed using a one-sided Mann–Whitney $U$ test (DEL-

enriched < DEL-neutral): $U = 7.183 \times 10^9$, $p = 1 \times 10^{-300}$ (normal approximation), effect size $r = -0.520$. **C** Monte Carlo–based estimation of enrichment factor comparing DEL-enriched and DEL-neutral targets through 1000 iterations of random resampling from the top 1% compounds, demonstrating the statistical robustness of enrichment trends. The line represents the mean enrichment factor, and the shaded area represents the mean ± standard deviation (SD) range. **D** Kernel density estimation (KDE) plots showing the distribution of docking scores for DEL-enriched (blue, $n = 16{,}507{,}848$ compounds) and DEL-neutral (red, $n = 16{,}448{,}758$ compounds) targets using a Gaussian kernel with bandwidth = 0.5.

resembling those that bind DEL molecules, suggesting that they are suitable for DEL screening. These potential targets cover a wide range of functional classes. In particular, seven protein classes were found to be highly enriched in both the DEL dataset and predicted human pockets, with transferases and hydrolases showing particularly high enrichment levels. Moreover, eleven additional human protein classes were also predicted to be suitable for DEL screening. These findings collectively demonstrate that the structural space amenable to DEL targeting is remarkably broad and diverse.

To clarify how DEL-compatible pockets relate to the traditional druggable space, we repeated our ErePOC cosine similarity analysis using FDA-AD reference pockets in place of DEL-derived ones. While both sets identified overlapping human pockets, enrichment patterns revealed distinct preferences. Among predicted high-similarity

pockets, FDA-AD-like pockets were predominantly enriched in classical target families such as receptors, ion channels, and isomerases, whereas DEL-like pockets were enriched in families including RNA-binding proteins, chromatin regulators, and GTPase activators (Fig. S18). Notably, transferases, hydrolases, and oxidoreductases were enriched by both predictions, highlighting shared accessible space. The relative scarcity of DEL-compatible receptor and channel targets may reflect the practical challenges of screening membrane proteins in DEL experiments. These results suggest that although DEL screening overlaps with the general druggable space, it exhibits a measurable bias toward target classes with larger, more hydrophobic, and chemically tractable binding pockets. For example, within the methionine adenosyltransferase family, the regulatory MAT2B harbors a larger, more hydrophobic pocket than the catalytic MAT2A (Fig. S19). Despite

MAT2A being a validated drug target[39], its pocket showed lower similarity to DEL reference pockets (cosine = 0.66) compared to MAT2B (cosine = 0.93), suggesting it may be less suitable for DEL screening. In silico DEL screening using docking confirmed that MAT2B exhibited a substantially more favorable mean docking score (−8.8 vs −5.3 kcal/mol). Together, these results support the ability of the ErePOC framework to identify targets with better DEL compatibility, enabling more efficient and strategically focused applications of DEL in early-stage drug discovery.

While ErePOC was developed to characterize and identify potential target proteins suitable for DEL screening, its representation of pocket functionality allows it to be applied to a wide range of downstream tasks, such as virtual screening[40,41], lipid-binding pocket prediction[42], reverse screening[43], molecule generation[44], and protein design[45]. In particular, when integrated with protein structure prediction frameworks such as AlphaFold3[46], ErePOC can significantly enhance the accuracy and utility of protein complex structure predictions, especially in optimizing and constraining pocket conformations. ErePOC may also facilitate the design of functional proteins with specific binding functionalities, which is a key advance in drug discovery and biomedical engineering[47,48]. The generalizability and adaptability of ErePOC underscore its potential as a versatile tool in AI-driven molecular design and target discovery.

## Methods

### Construction of ligand–pocket complexes datasets

Our study incorporates the following datasets (summarized in Table S1), which include both experimentally determined and computationally predicted structures of ligand–protein complexes, as well as datasets related to DELs and FDA-approved drugs. The first dataset consists of high-resolution experimental structures sourced from BioLiP2 (updated as of March 9, 2024), a semi-manually curated database focused on biologically relevant ligand–protein binding interactions. We extracted only regular ligands, excluding metals, peptides, DNA, and RNA, resulting in a final dataset of approximately 326,416 ligand–protein complexes. We specifically chose BioLiP2 rather than using all PDB entries with ligands because BioLiP2 systematically filters out crystallization artifacts and non-functional ligands, thereby ensuring greater biological relevance and reliability.

In BioLiP2, binding sites are originally defined based on functional residue annotations. Specifically, a protein residue is labeled as a binding residue if at least one of its atoms is within a distance equal to the sum of the van der Waals radii plus 0.5 Å of any atom of the bound ligand, and if there are at least two such inter-atomic contacts, the residue is considered part of the binding site. To ensure methodological consistency across datasets, we additionally defined binding pockets using a simple geometric criterion, including all protein atoms within 5 Å of any non-hydrogen atom of the bound ligand. All analyses were repeated using this definition, and the results remained consistent across both definitions (Figs. S8–S10 and S14–S15).

Additionally, we curated predicted ligand–protein complex structures from the AlphaFill database. AlphaFill transfers ligands, cofactors, and metal ions to AlphaFold-predicted models based on sequence and structural similarities. However, due to potential clashes between the transplanted ligands and the predicted receptor structures, the quality and resolution of the predicted complexes were relatively low. To improve the structural quality of AlphaFill entries, we implemented a steric clash detection procedure using an in-house Python script. Steric clashes were defined as any pair of non-hydrogen atoms separated by less than 1.2 Å. For each pocket and ligand, we calculated a clash score as the fraction of atom pairs that fell below this threshold. Protein-ligand clashes were quantified as the number of atom pairs between the pocket and ligand with inter-atomic distances

shorter than 1.2 Å. Structures were discarded if the pocket clash score was ≥0.005, or the ligand clash score was ≥0.01, or if the number of protein-ligand clashes was ≥5. This procedure removed approximately 11% of the AlphaFill dataset, yielding 293,019 relatively high-quality ligand–protein complexes for downstream analysis.

Our third dataset focused on FDA-approved drugs, collected from the DrugBank database. We gathered information on 2863 FDA-approved drugs and used their SMILES representations to search the Protein Data Bank (PDB) for corresponding drug–protein complexes. In total, 5286 potential drug–protein complex structures were identified. Ligands in these structures were marked using the "HET" keyword, and Tanimoto similarity based on Morgan fingerprints (radius = 2) was calculated using RDKit (version 2024.03.2) between these ligands and the FDA drug list. A ligand was considered an FDA-approved drug only if its Tanimoto similarity equaled 1.0, ensuring exact chemical identity. Pockets were defined as amino acid residues within 5 Å of these ligands, resulting in a final dataset of 340 unique crystal structures, each corresponding to a distinct FDA-approved (FDA-AD) drug–protein pair. Multiple targets of the same drug were retained, but redundant crystal structures of the same drug–target combination were removed.

The fourth dataset was specifically developed for DEL compounds. This dataset was compiled from the DELOpen platform [https://www.delopen.org/] by WuXi AppTec, with the latest update in April 2023. We analyzed target proteins and their binding residues using public references, and corresponding structures were downloaded from the PDB. For each target protein, only the DEL hit with the best binding affinity was retained. The 3D conformation of each ligand was generated using energy minimization in the MOE. Induced fit docking was then performed in MOE to predict the binding poses of the DEL hits with the target proteins. If the experimental structure of a DEL hit–target protein complex was available, it was included in the dataset instead. This dataset ultimately consisted of 128 unique DEL hit–target protein structures. The corresponding SMILES strings of these ligands are provided in Supplementary Table S2. To ensure methodological consistency, we note that redundancy was handled differently across datasets depending on their intended use. Specifically, the BioLiP2 and AlphaFill datasets intentionally retain multiple structures bound to the same ligand, thereby exposing the model to diverse conformational states and binding environments. In contrast, for the FDA-AD and DEL datasets, we applied stricter curation: the FDA-AD set was reduced to unique FDA drug–protein pairs (removing redundant crystal structures for the same drug target), while the DEL set was limited to one highest-affinity hit per target protein.

Finally, we downloaded 23,391 predicted human protein structures from the AlphaFold Protein Structure Database (*Homo sapiens* species). Each of these proteins was processed using the Fpocket tool to predict potential binding pockets. To analyze the enrichment of amino acids in our benchmark datasets, we referenced the ratios of the 20 types of amino acids that are enriched in the PDB database. These ratios were obtained from NIMBioS. (https://legacy.nimbios.org///-gross/bioed/webmodules/aminoacid.htm).

### Pocket physicochemical characterization and shape similarity analysis

We used the Fpocket 3.0[27] tool to extract pocket descriptors from experimental structures in BioLiP2, FDA-AD, and the DEL datasets. These descriptors include volume, number of alpha spheres, proportion of apolar alpha spheres, mean local hydrophobic density, density of alpha spheres, and proportion of polar atoms. These properties were calculated to provide insights into the physical characteristics of the binding pockets. Specifically, volume reflects the size of the cavity, with larger values indicating larger pockets. The number of alpha

spheres and the proportion of apolar alpha spheres capture the geometric and hydrophobic features of the pocket, with a higher proportion of apolar spheres suggesting a more hydrophobic environment. The mean local hydrophobic density serves as a descriptor for the overall hydrophobicity of the pocket, with higher values indicating greater hydrophobicity. The density of alpha spheres provides an indication of the packing density of the pocket, while the proportion of polar atoms helps assess the hydrophilicity of the binding site.

## Pocket–ligand interaction analysis using Arpeggio

We analyzed pocket–ligand interactions using the Arpeggio[28] tool, which identifies contacts at the pocket–ligand interface. These contacts were extracted from the ".bs_contactsfile" file generated by Arpeggio. The interaction types included in our analysis were van der Waals, hydrogen bonds, weak hydrogen bonds, halogen bonds, ionic interactions, aromatic interactions, hydrophobic interactions, carbonyl interactions, polar interactions (contacts between polar atoms that do not meet the geometric criteria for hydrogen bonds), and weak polar interactions. For each ligand–pocket complex, we summed the number of interactions of each type between the ligand and pocket atoms. The proportion of each interaction type was calculated by dividing its count by the total number of interactions, allowing us to compare the relative contributions of different interaction types to the overall binding profile.

## Statistical analysis and PCA projection

Nine pocket and interaction features were extracted using Fpocket and Arpeggio. The distributions of these features in the FDA-AD, DEL, and BioLiP2 datasets were first assessed for normality. As most features deviated from normality and the sample sizes were unequal, non-parametric statistical methods were employed. Specifically, the Kruskal–Wallis $H$ test was used to examine overall differences across datasets, followed by pairwise Mann–Whitney $U$ tests when significant, with Cliff's δ calculated to quantify effect sizes. All analyses were performed in Python using the SciPy library, and statistical significance was reported as $p$ values. To explore differences in the pocket feature space, BioLiP2 features were standardized (mean = 0, variance = 1) and subjected to PCA analysis. DEL and FDA-AD data were then projected onto the BioLiP2 PCA space, and scatter plots were generated to visually compare the distributions of the three datasets.

## Evaluation of molecular properties and ligand similarity

We evaluated the molecular properties of ligands from the BioLiP2 dataset, DEL hits, and FDA-approved drugs. The properties analyzed included the number of hydrophobic atoms (a_hyd), the number of hydrogen atoms (a_nH), the log octanol/water partition coefficient (logP), the log solubility in water (logS), and the topological polar surface area (TPSA). These properties were plotted to visualize their distribution across BioLiP2 ligands, FDA-approved drugs, and DEL hits. Ligand similarity was calculated using extended connectivity fingerprints (ECFP4), generated with the RDKit (version 2024.03.2) package[49], to compare the molecular structures and assess their similarities across the datasets.

## ErePOC model architecture

**Pocket featurization.** The input to the ErePOC model is the full-length amino acid sequence of the target protein. Each amino acid in the sequence is transformed into an embedding vector using the ESM-2 model. The embedding vector $\mathbf{E} \in \mathbb{R}^{d_t}$ has a default dimension of $d_t = 1280$ as defined by ESM-2. The binding pocket is represented by a vector $\mathbf{P} \in \mathbb{R}^{d_t}$, which is computed as the mean of the embedding vectors for the amino acids located within the binding pocket. This mean pooling process generates a feature vector that effectively captures the overall characteristics of the binding pocket.

**Mapping into a common latent space.** To compare pockets in a shared latent space, we first map the pocket embeddings $\mathbf{P_1} \in \mathbb{R}^{d_t}$ and $\mathbf{P_2} \in \mathbb{R}^{d_t}$ into a common latent space. This is achieved by applying a fully connected layer with GELU (Gaussian Error Linear Unit) activation, which transforms the pocket embeddings into latent vectors $\mathbf{T_1^*} \in \mathbb{R}^h$ and $\mathbf{T_2^*} \in \mathbb{R}^h$. The transformation is parameterized by weight matrices $\mathbf{W_t} \in \mathbb{R}^{h*d_t}$, and bias vectors $\mathbf{b_t} \in \mathbb{R}^h$, described by the following equation:

$$\mathbf{T}^* = \text{GELU}(\mathbf{W_t P} + \mathbf{b_t}) \tag{1}$$

Once we obtain the latent embeddings $\mathbf{T_1^*}$ and $\mathbf{T_2^*}$, their cosine similarity is computed to measure the distance between the two pockets:

$$\text{CosineSimilarity}(\mathbf{T_1^*}, \mathbf{T_2^*}) = \frac{\mathbf{T_1^*} \cdot \mathbf{T_2^*}}{\|\mathbf{T_1^*}\|_2 \cdot \|\mathbf{T_2^*}\|_2} \tag{2}$$

Although the pockets themselves are not directly annotated in our dataset, each pocket is associated with a specific binding ligand. To differentiate between pockets, we use the KL divergence as the loss function. The idea is that pockets whose ligands are similar—whether in structure, function, or chemical properties—should be represented closer together in the latent space. The loss function is defined as:

$$\text{loss} = P_{ij} \log\left(\frac{P_{ij}}{Q_{ij}}\right) \tag{3}$$

Here, $Q_{ij}$ represents the similarity between the ligands bound to pockets $i$ and $j$, computed using the Morgan fingerprint, which encodes each molecule into a fixed-dimensional embedding $\mathbf{M} \in \mathbb{R}^{d_m}$, where $d_m = 2048$. This formulation ensures that the model distinguishes between pockets by leveraging the similarity of their associated ligands, while guiding the pockets to be represented closely in the latent space if their ligands are similar.

**Implementation.** Model training was performed using the Adam optimizer with an initial learning rate set to 0.001. Early stopping with a patience of 100 epochs was applied, saving the model with the lowest validation loss. The maximum number of epochs was set to 2000. Each training epoch consisted of 4096 batches with 128 samples per batch, while validation used 256 batches of 128 samples. The model architecture included a 2-layer NN, with intermediate dimension $d = 512$ and output dimension $d = 256$. Each layer applied batch normalization, followed by dropout (rate = 0.1), and GELU activation. The model was implemented in PyTorch version 2.3.1 and trained on a machine equipped with NVIDIA L40 GPUs.

**Validation through binding pocket classification.** To assess the performance of ErePOC, we constructed a dataset of approximately 43,000 protein binding pockets, each labeled with one of seven types of bound ligands: ADP, COA, FAD, HEM, NAD, NADP, and SAM. We then designed a downstream classification task to evaluate the ability of our model to distinguish between these different binding pockets. For a comprehensive evaluation, we compared our model against three baseline models using zero-shot learning: ErePOC-NN, ESM-2-NN, ErePOC-SVM, and ESM-2-SVM. In ErePOC-NN and ErePOC-SVM, the pocket representations derived from contrastive learning were used as input features, with classifiers being NN or SVM, respectively. Similarly, in ESM-2-NN and ESM-2-SVM, the features directly extracted from the ESM-2 model were used, with NN and SVM classifiers employed for the classification task. The dataset was split 9:1 into training and test sets. Model performance was evaluated using precision, recall, and F1-

score:

$$Recall = \frac{TP}{TP + FN} \tag{4}$$

$$Precision = \frac{TP}{TP + FP} \tag{5}$$

$$F1 - score = 2 \times \frac{Precision \times Recall}{Precision + Recall} \tag{6}$$

where TP stands for True Positives, FP for False Positives, and FN for False Negatives.

## Hypergeometric test for enrichment analysis

In this study, we applied a hypergeometric test[50] to assess the enrichment of human proteins whose binding pockets exhibit high cosine similarity (greater than 0.8) to pockets known to bind compounds in DELs. The goal was to determine whether specific protein families or functional categories are significantly enriched within the subset of human proteins containing DEL-suitable pockets. For each human protein in the predicted dataset, we retrieved functional annotations using UniProt IDs and categorized them according to molecular functions and protein families, as defined in the UniProt database. This classification allowed us to categorize the human proteins into specific functional groups, such as enzyme families or cellular processes. We then calculated the enrichment score (ES) for each protein class, comparing its representation in the subset of DEL-suitable human proteins to its representation in the overall human proteome. Specifically, the enrichment score for a given protein class $i$ was defined as:

$$Enrichment\_score(ES)_i = \frac{N(i)_M}{N(i)_{total}} \Big/ \frac{N_M}{N_{total}} \tag{7}$$

where $N(i)_M$ is the number of proteins of the $i$th classification in the subset $M$, which includes the human proteins with pockets that exhibit cosine similarity >0.8 to known DEL pockets. $N(i)_{total}$ is the total number of proteins of the $i$th classification in the entire human protein dataset. $N_M$ and $N_{total}$ are the total numbers of proteins in subset $M$ and the entire dataset, respectively. An enrichment score greater than 1 indicates that the given protein class is overrepresented in the DEL-suitable subset. To evaluate statistical significance, we computed the $p$ value using the hypergeometric distribution:

$$p(observed) = \frac{\binom{N(i)_{total}}{N(i)_M} \binom{N_{total} - N(i)_{total}}{N_M - N(i)_M}}{\binom{N_{total}}{N_M}} \tag{8}$$

## Virtual DEL screening by docking

A virtual DEL comprising approximately 2.8 million small molecules was obtained from a publicly available dataset[36]. This dataset was originally developed to benchmark DEL-based hit discovery strategies and represents a chemically diverse and synthetically tractable compound space suitable for DEL screening. All compounds were downloaded in SMILES format and preprocessed using Open Babel (v3.1.1)[51]. Three-dimensional conformations were generated using Open Babel's OBBuilder module, with hydrogen atoms added to each molecule. Energy minimization was then performed using the MMFF94 force field. After preprocessing, a total of 2,778,654 unique DEL molecules were retained for virtual screening.

Docking was performed using AutoDock Vina (Version 2.0)[52]. Receptor and ligand structures were converted from PDB or MOL2 formats to PDBQT with MGLTools (v1.5.7). Receptor conformations were kept rigid, while ligand torsions were treated as flexible. The docking grid was defined as a 15 Å cubic box centered on the predicted DEL-compatible pockets with ErePOC. The default Vina scoring functions and search parameters were applied, with the exhaustiveness set to 8. For each ligand, the top five docking poses were generated, and the pose with the lowest binding energy was selected for subsequent analysis.

Docking scores for each compound were normalized using $Z$-score transformation:

$$Z_i = \frac{S_i - \mu_{all}}{\sigma_{all}} \tag{9}$$

where $S_i$ is the docking score of compounds $i$, and $\mu_{all}$ and $\sigma_{all}$ are the mean and standard deviation of docking scores. The enrichment factor (EF) at a given docking score cutoff $c$ was defined as:

$$EF(c) = \frac{N_{pos}(c)/N_{pos,\,total}}{N_{neg}(c)/N_{neg,\,total}} \tag{10}$$

where $N(c)$ and $N_{total}$ denote the number of compounds with docking scores $\leq c$ and the total number of compounds, respectively. The subscripts pos and neg refer to the DEL-enriched and DEL-neutral target sets.

To evaluate the statistical significance and robustness of the observed differences between DEL-enriched and DEL-neutral targets, the top 1% of docking scores per target were extracted and converted to $Z$-scores, and the distributions of these top 1% $Z$-scores were compared between groups using a one-sided Mann–Whitney $U$ test. In parallel, a Monte Carlo resampling procedure was applied to the same subsets, in which 200 compounds were randomly sampled from the combined pools over 1000 iterations. For each iteration, the EF was calculated across multiple docking score cutoffs, and the mean ± standard deviation across iterations was used to quantify the variability and stability of the EF values.

## Reporting summary

Further information on research design is available in the Nature Portfolio Reporting Summary linked to this article.

# Data availability

The datasets generated and analyzed in this study have been deposited in the Zenodo database under [https://doi.org/10.5281/zenodo.18033921]. The raw data supporting the findings of this study, including model outputs and processed datasets, are available from Zenodo under this accession code. Source data underlying all quantitative figures with manageable size are provided as individual Source data Excel files with this manuscript, including Figs. 1, 2, 5 and 7, and Supplementary Figs. S1–S4, S9, S12–S13, S15 and S17–S19. Due to the large scale of the datasets underlying the t-SNE visualizations (Fig. 4 and Supplementary Figs. S5–S11 and S14), individual data points are not provided as Source data files; however, the full processed inputs required to reproduce these figures are available on Zenodo. Raw data related to Figs. 1, 2, 7, S2, S4, S13, S17 and S19 are also provided on Zenodo. PyMOL session files (.pse) used for structural visualization in Fig. 6 and Supplementary Fig. S16 are available on Zenodo as well. The BioLiP2, AlphaFill, and AF2-predicted protein structure data used in this study are publicly available from the BioLiP, AlphaFill, and AlphaFold databases at [https://zhanggroup.org/BioLiP/], [https://alphafill.eu/], and [https://alphafold.ebi.ac.uk/download], respectively. Lists of target proteins for BioLiP2, AlphaFill, and AlphaFold-predicted human proteins, as well as PDB code mappings for DEL and FDA-AD targets, are also available on Zenodo. Source data are provided with this paper.

## Code availability

All custom source code and algorithms generated in this study are publicly available at https://github.com/JingHuangLab/ErePOC, along with the processed data required to reproduce all results reported in this manuscript. There are no restrictions on access.

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

## Acknowledgements

This work was supported by the "Pioneer" and "Leading Goose" R&D Program of Zhejiang, grant numbers 2023C03109 (J.H.) and 2024SSYS0036 (J.H.); the National Natural Science Foundation of China, grant 32171247 (J.H.), T2596084 (J.H.), and 32501101 (W.Z.); the Zhejiang Provincial Natural Science Foundation, grant LQ23F020011 (W.Z.); the State Key Laboratory of Gene Expression; and the Westlake Education Foundation. The authors thank the Westlake University Supercomputer Center for computational resources and related assistance.

## Author contributions

J.H. and Q.H. conceived the study. W.Z., Y.W., R.Z., and R.Q. designed the experiments. All authors analyzed the results. J.H. and W.Z. wrote the manuscript.

## Competing interests

The authors declare no competing interests.
