## [Transparent Peer Review file · Nature Communications]

Deciphering DEL Pocket Patterns through Contrastive Learning

Corresponding Author: Dr Jing Huang

Version 0:

Reviewer comments:

Reviewer #1

(Remarks to the Author)

Noteworthy results:

The authors have developed a method, ErePOC, for creating ligand binding pocket representations and assess their functional similarity. They incorporated their embeddings into the ESM-2 contrastive learning method. They applied their representation method to a set of experimentally determined protein-ligand structures in the BioLiP2 database. The BioLiP2 database is a semi-curated set of proteins bound to native ligands. They demonstrated the ability of their method to accurately group pockets that bound the same ligand. They compared this method to performance of ESM-2 embeddings which were not able to distinguish between binding pocket classes. They applied their method to a set of experimentally derived protein-ligand structures from the OpenDEL database. The OpenDEL database contains experimental protein structures complexed to DEL hits. The resulting DEL pocket representations were applied to a filtered set of structures from the AlphaFold2 database of predicted protein structures that contained binding pockets identified using F-pocket. They identified proteins from the AlphaFold database that have binding pockets similar to those known to bind DEL molecules. They propose that this method can be used to identify targets suitable for DEL screening, virtual screening, used to identify binding pockets for a particular ligand type, and optimize binding pockets for a specific ligand.

Significance:

The method they have developed for binding pocket representations outperforms ESM-2 in its ability to group similar binding pockets. Generalizability based on test set performance was equally good for ErePOC-NN, ErePOC-SVM and ESM-2-NN models. While the analyses comparing the properties of pockets calculated using F-pocket are interesting, they do not represent a significant advancement to the field.

Additional experiments:

They identify pockets that they expect to bind DEL molecules but do not provide any experimental validation that their predictions are accurate. I suggest that the authors choose representative proteins from each of the 7 highly enriched DEL protein classes (Fig. 5) and perform a DEL screen. As a control pick representatives from 7 protein classes that were not highly enriched. If their predictions are correct, they should observe a significantly higher number of enriched DEL molecules in the positive set relative to the negative set.

Flaw in data analysis and interpretation:

On page 5, line 118, they describe how they defined binding pockets from the different datasets. For the BioLiP2 dataset, they used functional annotations provided by the database curation. For the FDA, DEL and AlphaFill datasets, they define the binding pocket by distance from the bound ligand. They don't explain why they used different criteria for defining the binding pockets.

Page 6, line 124 states "Drug candidates, being larger and chemically more complex than natural ligands, are specifically designed to exhibit stronger affinity." The authors do not cite a reference or provide an analysis to demonstrate that the statement is true. They should remove that statement or provide evidence.

Page 6, line 131 states that “DEL pockets were larger than FDA-AD ones, which aligns with the general understanding that DEL molecules are larger, with average molecular weights of 1298.5 Da and 218.3 Da in our datasets, respectively.” When evaluating the graphs in Figure S2, the molecular weights of the DEL molecules are higher than those of the FDA-AD molecules but the average MW of the DEL molecules appears to be much lower based on the distribution shown. The majority of the DEL molecules are in the 300-700 Da MW range. This is also inconsistent with the data analysis shown in Fig. 2A & B indicating that the volume of binding pockets is similar between the FDA-AD and DEL datasets.

Page 7, line 170 states “Overall, these results indicate that the binding of DEL hit molecules relies more on pocket shape complementarity and can be further optimized towards drug molecules.” I don’t follow how they made that conclusion based on the analysis and results described in section 2.2 and shown in Fig. 2.

Page 11, line 275 “The ErePOC representation partitions the BioLiP2 pocket space into distinct patterns, offering critical insights into the global pocket landscape. For example, pockets bound to native ligands such as SAM or HEM appear less suitable as drug targets - particularly in the context of DEL screening. In contrast, pockets bound to ADP seem more flexible to accommodate designed molecule.” These interpretations appear to be based on the analyses shown in Fig. C and D. While it is true that the chemical space corresponding to SAM or HEM bound structures lacks representatives from the DEL and FDA datasets, the chemical space corresponding to ADP bound structures has very few representatives from the DEL and FDA datasets.

Based on the description of database curation described in section 4.1, it isn’t clear to me if datasets were consistently constructed. The BioLiP2 dataset has multiple structures that are bound to the same ligand. The description of the process used for the AlphaFill structures given on page 16 lines 444-450 doesn’t indicate if the 330,434 are all unique or have redundant/similar ligands. The set of FDA molecules is described on page 17, line 458 as being 630 unique crystal structures. Does that mean they are 630 unique ligand-protein combinations? Page 17 line 462 “For each target protein, only the DEL hit with the best binding affinity was retained.” resulting in 128 unique structures.

Comments:

Page 2, line 15 states that “few DEL-derived compounds have advanced to clinical trials or reached the market.” I disagree with that assertion. I would refer the authors to a nice review written by David Liu in which he performed a comprehensive analysis of DEL derived molecules that have entered the clinic. Nat Rev Drug Discov. 2023 Jun 16;22(9):699–722. doi:10.1038/s41573-023-00713-6. While the total number of DEL molecules that have reached the clinic is comparatively low relative to molecules identified by high-throughput screening, DEL has only recently been adopted as a standard approach for hit identification.

The authors included structures from three databases of experimentally derived structures, BioLiP2, FDA and OpenDEL, and two databases of predicted structures, AlphaFill and AlphaFold. It would be interesting to know why they didn’t include in their comparisons the human protein structures from the Protein Data Bank with small molecule ligands not included in the BioLiP2, FDA and OpenDEL databases.

Section 2.5 described the identification of pockets from the human proteome that are suitable for DEL screening. The set of AlphaFold2 predicted human protein structures was filtered prior to analysis with ErePOC2. One of the filters applied was to remove proteins that had pocket volumes less than 800 Å³. On page 7, line 151 they indicate that the average volume of a DEL molecule binding pocket is 2723.3 Å³. They should explain why they decided to use 800 Å³ as their volume cutoff.

Figs. 2A and S10A are lacking units for the X-axis.

Figs. 2C and S10C the legend below the graph has a typo: “polor” should be “polar”

Fig. S3 should include a legend indicating the corresponding binding sites and colors as was done in Fig. 4.

(Remarks on code availability)

Reviewer #2

(Remarks to the Author)

The authors describe a method to represent the properties of protein pockets that captures both structural and functional features, and use this method to investigate DEL target space.

The paper is well written, but we have a number of comments around the methodology.

Comments:

The DEL dataset contains 128 entries. It would be useful if the authors included the associated SMILES strings for the ligands and a depiction of their structures (as supplementary information).

“We hypothesize that DEL targets exhibit unique pocket characteristics that can be utilized in drug design decision-making, such as target prioritization and hit identification.” Does this study shed any light on these unique pocket characteristics? Or indeed is the conclusion that the pocket characteristics are similar to general druggable pocket?

P6, L123: “DEL and FDA-AD pockets were notably larger than those in BioLiP2 and AlphaFill, likely due to differences in

ligand characteristics." I do not believe that this shows that the pockets are larger, merely that the ligands are larger, and hence the number of residues within 5Å. The pocket size is accurately assessed in Section 2.2.

P7: Across the various properties, is the difference between DEL and FDA-AD statistically significant? Is there any evidence that the targets for which DEL has been run differ from the set of all druggable targets?

The FDA-AD compounds are the result of a lead optimisation process that would have eliminated functional groups that are problematic for metabolism. The DEL compounds are investigational compounds and their structures are expected to be less optimal. This might explain why ionic interactions were found in the DEL dataset. If the authors had included the associated ligand structures for the DEL dataset, this would be easier to assess.

The phrase "DEL pockets" seems a bit misleading. These are just pockets where DEL ligands have been reported.

Presumably, small molecules have also been reported for these pockets, or is this not the case?

P 7, L174: "DEL pockets possess unique features in their physicochemical properties, distinguishing them from general protein pockets." The results shown seem to indicate that their properties are similar to those for FDA-AD.

Fig 5: If prediction of human protein targets suitable for DEL screening had been done instead with the Drug-AD pockets would there have been any difference in the results? If so, this would be interesting to report. What I am trying to figure out is whether the 'DEL-able' pocket space really is different than general druggable pocket space. On P11, it is stated that "Together, these results suggest that DEL screening can readily access most of the known druggable target space." which would indicate that there is little difference.

P. 13: "These examples highlight how contrastive learning imparts the capacity to detect higher-level functional or physicochemical similarities that are not solely reflected by either global or local 3D structural alignment." Are these not false positives, or is there experimental evidence that these bind very similar ligands?

P17: "A ligand was considered an FDA-approved drug if its Tanimoto similarity exceeded 0.7." Unless the similarity is 1.0, the structure is different. It may be a tautomer, in which case you could use the InChI to check for identity. But otherwise, these will not be the same molecule, and I believe that it is incorrect to state that these are FDA-approved drugs. They simply have high similarity to FDA-approved drugs.

P5, line 112: "i) the 112 BioLiP2(22) database, which provides experimentally determined binding structures of natural ligands and their associated pockets."

This sentence is confusing, as BioLip2 is not limited to natural ligands. If authors used a subset of BioLip2, it should be clearly stated here. Method section, line 438 mentions the filter for "regular ligands", but this is not clear in the previous sections.

P5, line 118: "They include 326,416, 330,434, 128, and 630 pockets, respectively"

The study's comparative analysis is challenged by the data imbalance, with the BioLip2 and AlphaFill datasets being orders of magnitude larger than the primary DEL dataset (n=128). Could the authors comment on this and whether it may effect the statistical power of the conclusion?

P5, 118-123: For the BioLiP2 dataset, pockets are defined functionally based on "experimentally annotated binding residues," which typically includes a small set of specific interacting residues. For all other datasets, pockets are defined geometrically using a universal 5Å distance cutoff from the ligand which creates a more general, inclusive cavity. This inconsistency introduces a systematic bias in the average number of residues observed in Figure 1A.

P7, line 164: "We further analysed the interactions between pocket residues and ligands using the Arpeggio method(25) (Figure 2 a-c). Hydrophobic interactions accounted for 50.7% in DEL, 49.5% in FDA-AD, and 30.1% in BioLiP2. Hydrogen bond interactions were 3.8% for DEL, 4.8% for FDA-AD, and 9.9% for BioLiP2, while polar interactions constituted 6.0%, 7.8%, and 14.8%, respectively. Ionic interactions can be found in the DEL dataset (1.50%) but were almost absent in FDA-AD (0.15%)."

The authors report "Hydrogen bond interactions" (e.g., 9.9% in BioLiP2) and "polar interactions" (14.8% in BioLiP2) as two distinct categories. This is confusing because a hydrogen bond is a specific and strong type of polar interaction. What does the "polar interactions" category actually contain if it excludes hydrogen bonds?

Moreover, the authors state that ionic interactions are more common in the DEL set (1.50%) than the FDA-AD set (0.15%). Given the tiny size of the DEL dataset (n=128), could this 1.5% be an artifact arising from just a few charged ligands in their small sample? Also, the corresponding value for the massive BioLiP2 dataset should be reported?

P16, line 446: The low-quality dataset from AlphaFill is a critical issue that is not adequately addressed. The authors' only acknowledgement of this problem is a single sentence stating that the "quality and resolution of the predicted complexes were relatively low" due to "potential clashes." It appears that the authors used all 330,434 structures without any quality-based filtering. Surely this brings into question the results from the AlphaFill analysis as the defined pockets may be artifacts or the wrong size.

P16, line 454-457: "Ligands in these structures were marked using the 'HET' keyword, and Tanimoto similarity based on molecular fingerprints was calculated 455 between these ligands and the FDA drug list. A ligand was considered an FDA-approved drug if its 456 Tanimoto similarity exceeded 0.7."

What fingerprint (FP) was used when doing the Tanimoto calculation? Without knowing this, the score thresholds are meaningless as interpretation varies from one FP to another. In any case, 0.7 seems very low, and it may find similar

compounds but not the exact molecule.

P17, line 458: "...resulting in a final dataset of 630 unique crystal structures of drug-pocket complexes." Although authors mention unique crystal structures, they fail to address the massive overrepresentation of certain drug-target families in the Protein Data Bank (PDB) (e.g., kinases, HIV protease, and heat shock proteins). Dozens or even hundreds of these PDBs could contain the exact same drug binding to the exact same protein pocket. The results will reflect the specific properties of a few over-studied protein families rather than the general characteristics of FDA-approved drug pockets.

(Remarks on code availability)

I note that the code is on GitHub and includes instructions on setting it up with Conda. Unfortunately, my institution does not permit access to Conda due to its new license and so I have not proceeded further.

Reviewer #3

(Remarks to the Author)

(Remarks on code availability)

Version 1:

Reviewer comments:

Reviewer #2

(Remarks to the Author)

The authors have addressed the issues we raised.

Signed Noel M. O'Boyle, Melissa F. Adasme

(Remarks on code availability)

Reviewer #3

(Remarks to the Author)

(Remarks on code availability)

We would like to thank all reviewers for their careful readings and comments. We are very glad to work with them to improve the manuscript.

Response to Reviewer #1

1. The Reviewer commented:

Noteworthy results:

The authors have developed a method, ErePOC, for creating ligand binding pocket representations and assess their functional similarity. They incorporated their embeddings into the ESM-2 contrastive learning method. They applied their representation method to a set of experimentally determined protein-ligand structures in the BioLiP2 database. The BioLiP2 database is a semi-curated set of proteins bound to native ligands. They demonstrated the ability of their method to accurately group pockets that bound the same ligand. They compared this method to performance of ESM-2 embeddings which were not able to distinguish between binding pocket classes. They applied their method to a set of experimentally derived protein-ligand structures from the OpenDEL database. The OpenDEL database contains experimental protein structures complexed to DEL hits. The resulting DEL pocket representations were applied to a filtered set of structures from the AlphaFold2 database of predicted protein structures that contained binding pockets identified using F-pocket. They identified proteins from the AlphaFold database that have binding pockets similar to those known to bind DEL molecules. They propose that this method can be used to identify targets suitable for DEL screening, virtual screening, used to identify binding pockets for a particular ligand type, and optimize binding pockets for a specific ligand.

Significance:

The method they have developed for binding pocket representations outperforms ESM-2 in its ability to group similar binding pockets. Generalizability based on test set performance was equally good for ErePOC-NN, ErePOC-SVM and ESM-2-NN models. While the analyses comparing the properties of pockets calculated using F-pocket are interesting, they do not represent a significant advancement to the field.

Author reply:

We thank the reviewer for her or his constructive assessment and for recognizing the importance of pocket representation. The central contribution of this work is the development of ErePOC, a contrastive learning framework that generates ligand-aware, functional embeddings of protein pockets. Inspired by the specific challenge of assessing target suitability for DEL screening, ErePOC enables scalable comparison of pocket similarity across large structural datasets — and serves more broadly as a general tool for functionally informed pocket analysis.

While DELs have revolutionized early-phase drug discovery by enabling the screening of billions of compounds, the lack of systematic, structure-based methods for evaluating DEL-target compatibility remains a major bottleneck. Our study bridges this gap by combining protein language models with ligand similarity-driven training, resulting in an approach that

outperforms raw ESM-2 features and structure-based alignments in grouping functionally related pockets. The ErePOC framework allows us to expand the landscape of predicted DEL-suitable targets across the human proteome and uncover physicochemical and functional patterns that are not accessible by traditional descriptors alone.

The Fpocket-based analyses are not presented as a standalone contribution but to provide physicochemical context for describing pocket properties. These analyses serve both to motivate the development of a latent-space representation and to offer interpretability and cross-referencing for the results learned by ErePOC. To avoid overemphasizing these descriptive analyses, we have added the following sentence in the first paragraph of “Conclusion and Discussion” section to clarify this point.

“These insights motivate key innovation of this study: developing ErePOC to learn high-dimensional pocket representations beyond descriptive features, and applying it to computationally expand the landscape of DEL-suitable targets across the human proteome.”

We have also added the following sentences at the end of Section 2.2 “Analysis of pocket physicochemical properties and their interactions with ligands”:

“We note that no single feature or simple combination could distinguish DEL from FDA-AD or general protein pockets, possibly due to the broad variability of pocket structures. This observation highlights the need for developing a more informative representation of pockets.”

2. The Reviewer commented:

Additional experiments:

They identify pockets that they expect to bind DEL molecules but do not provide any experimental validation that their predictions are accurate. I suggest that the authors choose representative proteins from each of the 7 highly enriched DEL protein classes (Fig. 5) and perform a DEL screen. As a control pick representatives from 7 protein classes that were not highly enriched. If their predictions are correct, they should observe a significantly higher number of enriched DEL molecules in the positive set relative to the negative set.

Author reply:

We greatly appreciate the reviewer’s insightful suggestion. We fully agree that performing DEL screening on representative targets would provide valuable validation and further insights into the DEL compatibility of different protein classes. However, conducting such large-scale DEL experiments for 14 targets from a variety of protein classes would require substantial resources, including protein expression and purification, DEL screening, sequencing, and downstream analysis, which lie well beyond the capabilities of most, if not all, academic laboratories, and the scope of this work. The primary goal of our work is to develop a computational framework that can systematically identify protein targets potentially more suitable for DEL screening, helping to prioritize targets and reduce experimental burden in early-stage drug discovery.

As a computational substitute, we designed a virtual DEL screening workflow for 14 representative protein targets to emulate experimental testing. Specifically, we leveraged a DEL compound library containing approximately 2.8 million DEL compounds and performed structure-based docking calculations against predicted DEL-suitable targets and matched control targets from non-enriched protein classes.

Although docking cannot fully recapitulate experimental outcomes, previous studies have shown that large-scale docking trends often correlate with hit discovery rates. Consistent with this, we observed that DEL-enriched targets yielded significantly more favorable docking scores, higher enrichment of top-ranking compounds, and more stable Monte Carlo enrichment profiles compared to DEL-neutral controls. Inspired by the suggestion, we also discussed a few particular cases, such as MAT2A and MAT2B. These results, now included in the revised manuscript (Section 2.6, Figures 7 and S16-S17), provide computational evidence that supports the predictive utility of ErePOC and align with the validation envisioned in the reviewer's suggestion.

“As a further validation, we designed a large-scale in silico DEL screening experiment against 14 selected human protein targets to compare their capability to bind DEL-like compounds (Figure S16). Six of these targets, each drawn from a different DEL-enriched functional family identified in our earlier analysis, contained a pocket exhibiting high ErePOC cosine similarity (> 0.8) to known DEL pockets. These include a chromatin regulator (O15294), a hydrolase (P03951), an isomerase (P26885), a lyase (P43166), a multifunctional enzyme (P14060), and an RNA-binding protein (Q8TAS1). As a comparison group, six additional targets were selected from DEL-neutral protein families, which did not show enrichment but nevertheless contained pockets with high ErePOC similarity (> 0.8). These include a signal transduction inhibitor (O14508), a mitogen (Q9H706), an elongation factor (P43897), an actin capping protein (P47756), a hypotensive agent (P68871), and a cyclin (Q5T5M9). In addition, two closely related proteins, MAT2A (P31153) and MAT2B (Q9NZL9), were included as a family-level case study. The catalytic MAT2A contains a small, polar pocket with a maximum ErePOC similarity of only 0.66 to known DEL pockets. In contrast, the regulatory MAT2B, though non-catalytic, harbors a larger and more hydrophobic pocket with a much higher similarity (0.93), providing an illustrative example of differential DEL compatibility within a single protein family.

Docking calculations were performed using a published virtual DEL compound library (approximately 2.8 million molecules) sampled from 15 sublibraries of the three-cycle HitGen OpenDEL library³⁶. No DNA tags were added so they represent off-bead synthesized molecules. Motivated by previous studies showing that more favorable (i.e., more negative) docking scores correlate with higher experimental hit rates, and that larger libraries improve discovery efficiency^{37, 38}, we evaluated whether ErePOC-identified pockets exhibit similarly favorable trends for DEL compounds. Standardized docking Z-score distributions, calculated using the global mean and standard deviation of docking scores across all compounds, are shown in Figure 7A. Targets from DEL-enriched families exhibited stronger predicted binding, with more negative mean Z-scores (−2.18 vs −1.07) and docking scores (−7.45 vs −6.15 kcal mol^{−1}) compared to those from DEL-neutral families. The difference was also statistically significant when examining the top 1% scoring compounds per target: DEL-enriched targets yielded mean top 1% docking scores ranging from −8.93 to −11.96 kcal mol^{−1} (Z = −1.54 to −3.73), whereas DEL-

neutral targets ranged from -5.49 to -9.73 kcal mol⁻¹ ($Z = +0.95$ to -2.12) (Figure 7B). Monte Carlo resampling over 1,000 iterations (Figure 7C) confirmed the robustness of these trends across relevant cutoffs (-10 to -8 kcal mol⁻¹), supporting a significant difference in the predicted ability to bind DEL molecules as measure by docking scores. These differences were also evident in the distribution of docking scores (Figure 7D and Supplementary Figure S17).” (Results in Section 2.6)

Figure 7. Docking score comparison and enrichment analysis of DEL-enriched and DEL-neutral targets. **A.** Smoothed cumulative distribution functions (CDFs) of standardized docking Z-scores, illustrating a shift toward more favorable binding scores for DEL-enriched targets. **B.** Mann–Whitney U test comparing top 1% docking Z-scores between DEL-enriched and DEL-neutral groups. Stars represent the mean Z-score of the top 1% compounds for each target. **C.** Monte Carlo–based estimation of enrichment factor (EF) comparing DEL-enriched and DEL-neutral targets through 1000 iterations of random resampling from the top 1% compounds, demonstrating the statistical robustness of enrichment trends. **D.** Kernel density estimation (KDE) plots showing the distribution of docking scores for DEL-enriched (blue) and DEL-neutral (red) targets.

Figure S16. Selected human proteins and their pockets for in silico DEL screening. This figure shows the 14 human protein targets selected for large-scale virtual screening on a DEL compound library, grouped into DEL-enriched (left panels) and DEL-neutral (right panels) categories based on computational predictions. The MAT2B and MAT2A (bottom panels) are included as an example for different predicted DEL compatibility within a single protein family. These classifications support the experimental design used to evaluate ErePOC-predicted DEL suitability.

Figure S17. Docking score distributions for the 14 human protein targets.

3. The Reviewer commented:

Flaw in data analysis and interpretation:

On page 5, line 118, they describe how they defined binding pockets from the different datasets. For the BioLiP2 dataset, they used functional annotations provided by the database curation. For the FDA, DEL and AlphaFill datasets, they define the binding pocket by distance from the bound ligand. They don't explain why they used different criteria for defining the binding pockets.

Author reply:

We sincerely thank the reviewer for raising this important concern. In our study, binding pockets in the FDA-AD, DEL, and AlphaFill datasets were consistently defined using a 5 Å distance cutoff from ligand atoms. The BioLiP2 dataset is annotated with binding pockets, so we originally used the curated functional annotations provided by the database. To ensure methodological consistency and assess robustness, we additionally redefined BioLiP2 pockets using the same 5 Å geometric criterion applied to the other datasets. This unified definition was used to re-analyze all datasets and consistent results were obtained between these two BioLiP2 pocket definition.

In the revised manuscript (Results, Section 2.1), we now clarify both definition:

“For BioLiP2, pockets were defined using experimentally annotated binding residues provided by the webserver. Meanwhile, pockets for the AlphaFill, DEL, and FDA-AD datasets were generated by including all amino acid residues within 5 Å of the bound ligand. To assess the impact of this inconsistency, we additionally redefined BioLiP2 pockets using the same distance-based criterion and repeated all analyses under this unified definition. The key findings remained consistent across definitions, suggesting that our conclusions are reasonably robust to differences in pocket definition.”

We have also added the following details in the Methods section (4.1):

“In BioLiP2, binding sites are originally defined based on functional residue annotations. Specifically, a protein residue is labeled as a binding residue if at least one of its atoms is within a distance equal to the sum of the van der Waals radii plus 0.5 Å of any atom of the bound ligand, and if there are at least two such inter-atomic contacts, the residue is considered part of the binding site. To ensure methodological consistency across datasets, we additionally defined binding pockets using a simple geometric criterion, including all protein atoms within 5 Å of any non-hydrogen atom of the bound ligand. All analyses were repeated using this definition, and the results remained consistent across both definitions (Figs. S8–S10 and S14–S15).”

We re-ran all analyses using the redefined pockets. Under this unified definition, the average pocket sizes are: BioLiP2: 12.5 residues, AlphaFill: 12.5, DEL: 28.1, and FDA-AD: 16.1. As expected, DEL and FDA-AD pockets remain larger than BioLiP2 and AlphaFill, reflecting the generally larger and more chemically complex nature of synthetic drug molecules and DEL hits compared to natural ligands. While minor quantitative changes were observed (e.g., slightly

lower enrichment of charged residues and slightly higher hydrophobic residue proportions in BioLiP2), the overall trends and conclusions remain unchanged: DEL pockets are larger and more hydrophobic, exhibiting weaker polar and hydrogen-bond interactions than BioLiP2 pockets. We have also tested the robustness of ErePOC against these two slightly different pocket definitions, and found very consistent results. Accordingly, we have added the following sentences to the Section 2.3:

“We note that for ErePOC model training and downstream analyses, the original BioLiP2 binding residue annotations were used to preserve experimentally validated contacts. Training and analyses using the alternative 5 Å pocket definition yielded highly similar results, as demonstrated in Figure S8-S10.”

Figure S8. Cosine similarity comparison across different binding pockets under two BioLiP2 pocket definitions. A. Cosine similarity results based on the original BioLiP2 annotation-defined pockets, including ESM-2 vectors, t-SNE 2D representations after ErePOC transformation, and ErePOC embeddings; **B.** Corresponding results obtained using the unified 5 Å pocket definition. The overall similarity distributions remain consistent across the two definitions, confirming the robustness of ErePOC embeddings with respect to pocket boundary choice.

Figure S9. Comparative performance of four models under two BioLiP2 pocket definitions. A. Predictive performance of ErePOC-NN, ErePOC-SVM, ESM2-NN, and ESM2-SVM using BioLiP2 annotation-defined pockets; **B.** Corresponding results using the unified 5 Å pocket definition.

Figure S10. t-SNE visualization of ErePOC and ESM-2 representations for BioLiP2 dataset using the 5 Å pocket definition. A–B. Visualization of the seven types of ligand-binding pocket landscapes using ESM-2 (A) and ErePOC (B) representations, respectively. **C.** Pocket landscape using ErePOC representations. **D.** Comparison of pocket landscapes for the FDA-AD and DEL datasets against BioLiP2 using ErePOC representations. Results are consistent with those obtained using curated BioLiP2 binding annotations (Figure 4), confirming the robustness of the ErePOC embedding with respect to pocket definition.

4. The Reviewer commented:

Page 6, line 124 states “Drug candidates, being larger and chemically more complex than natural ligands, are specifically designed to exhibit stronger affinity.” The authors do not cite a reference or provide an analysis to demonstrate that the statement is true. They should remove that statement or provide evidence.

Author reply:

We thank the reviewer for this helpful comment. We have cited the following two references to support the statement and revised the sentence for improved clarity:

Leeson PD, Springthorpe B. The influence of drug-like concepts on decision-making in medicinal chemistry. *Nat Rev Drug Discov* 6, 881-890 (2007).

Bickerton GR, Paolini GV, Besnard J, Muresan S, Hopkins AL. Quantifying the chemical beauty of drugs. *Nat Chem* 4, 90-98 (2012).

The revised sentence now reads:

“Synthetic drug molecules are often optimized for potency and selectivity through medicinal chemistry efforts, which typically results in larger and more chemically complex scaffolds compared to regular ligands^{25, 26}. They often target protein pockets that are more spacious, flexible, and dynamic, accommodating a broader range of interactions.” (Section 2.1)

5. The Reviewer commented:

Page 6, line 131 states that “DEL pockets were larger than FDA-AD ones, which aligns with the general understanding that DEL molecules are larger, with average molecular weights of 1298.5 Da and 218.3 Da in our datasets, respectively.” When evaluating the graphs in Figure S2, the molecular weights of the DEL molecules are higher than those of the FDA-AD molecules but the average MW of the DEL molecules appears to be much lower based on the distribution shown. The majority of the DEL molecules are in the 300-700 Da MW range. This is also inconsistent with the data analysis shown in Fig. 2A & B indicating that the volume of binding pockets is similar between the FDA-AD and DEL datasets.

Author reply :

We thank the reviewer for carefully identifying this mistake. The originally reported molecular

weights were incorrect due to a data processing error. We have now corrected the average molecular weights to 560.5 Da for DEL molecules and 310.9 Da for FDA-AD molecules in the revised manuscript. The text on page 6, line 131 has been updated as follows.

“The average molecular weights of DEL and FDA-AD ligands in this study are 560.5 and 310.9 Da, respectively (Figure S2).” (Section 2.1)

6. The Reviewer commented:

Page 7, line 170 states “Overall, these results indicate that the binding of DEL hit molecules relies more on pocket shape complementarity and can be further optimized towards drug molecules.” I don’t follow how they made that conclusion based on the analysis and results described in section 2.2 and shown in Fig. 2.

Author reply:

We thank the reviewer for this helpful comment. In the revised manuscript, we have clarified the reasoning and reworded the statement to avoid misinterpretation.

Our conclusion is based on two main observations:

- DEL pockets tend to be larger than BioLiP2 pockets and modestly larger than FDA-AD pockets based on pocket volume and alpha sphere analyses (Figure S3A, Cliff’s δ), suggesting a more expansive surface area for ligand–protein interactions, which is a geometric prerequisite for shape complementarity–driven binding.
- Interaction analysis (Figure S3B) shows that hydrophobic contacts dominate in DEL pockets, while hydrogen bonds and polar interactions are less frequent. This profile is consistent with a binding mode that relies more on nonspecific shape and hydrophobic complementarity rather than defined polar interactions.

To ensure the interpretation remains appropriately cautious and logically supported, we have revised the text in the manuscript as follows:

“Overall, these results indicate that DEL hits preferentially bind to larger, more hydrophobic pockets. The expansive contact area in these pockets enhances binding through shape complementarity, thereby favoring hydrophobic interactions. These characteristics suggest potential avenues for optimization toward drug-like molecules, for example by balancing polar interactions to improve binding specificity.” (Section 2.2)

7. The Reviewer commented:

Page 11, line 275 “The ErePOC representation partitions the BioLiP2 pocket space into distinct patterns, offering critical insights into the global pocket landscape. For example, pockets bound to native ligands such as SAM or HEM appear less suitable as drug targets - particularly in the context of DEL screening. In contrast, pockets bound to ADP seem more flexible to accommodate designed molecule.” These interpretations appear to be based on the analyses shown in Fig. C and D. While it is true that the chemical space corresponding to SAM or HEM bound structures lacks representatives from the DEL and FDA datasets, the chemical space corresponding to ADP bound structures has very few representatives from

the DEL and FDA datasets.

Author reply:

We thank the reviewer for his or her careful reading and thoughtful reasoning. We agree that the interpretation regarding ADP-bound pockets may have been overstated, given the limited number of DEL and FDA-AD representatives in that region of the pocket space. We have thus removed this specific interpretation. We have retained the discussion of SAM/HEM-bound pockets, emphasizing that their absence from DEL and FDA-AD chemical spaces suggests that these tightly bound, cofactor-associated pockets may be less suitable for conventional DEL screening. The revised text now reads as follows:

“The ErePOC representation partitions the BioLiP2 pocket space into distinct patterns, offering critical insights into the global pocket landscape. For example, pockets bound to native ligands such as SAM or HEM are notably absent from the DEL and FDA-AD chemical spaces, suggesting that these tightly bound, cofactor-associated pockets may be less suitable for conventional DEL screening. To further explore the “DEL-able” regions of the pocket landscape, we identified the five nearest neighbors for each DEL target in the BioLiP2 dataset based on cosine similarity (Figure S12).” (Section 2.4)

8. The Reviewer commented:

Based on the description of database curation described in section 4.1, it isn't clear to me if datasets were consistently constructed. The BioLiP2 dataset has multiple structures that are bound to the same ligand. The description of the process used for the AlphaFill structures given on page 16 lines 444-450 doesn't indicate if the 330,434 are all unique or have redundant/similar ligands. The set of FDA molecules is described on page 17, line 458 as being 630 unique crystal structures. Does that mean they are 630 unique ligand-protein combinations? Page 17 line 462 “For each target protein, only the DEL hit with the best binding affinity was retained.” resulting in 128 unique structures.

Author reply:

We thank the reviewer for this helpful comment regarding dataset consistency and construction. We have revised Section 4.1 to clarify these issues. Specifically:

- BioLiP2 dataset: Multiple structures bound to the same ligand are retained intentionally, allowing ErePOC to learn from diverse conformational states of the same ligand across different proteins and crystallization conditions. This enhances functional relevance and improves generalizability.
- AlphaFill dataset: Similarly, AlphaFill contains redundant ligand–protein complexes. Redundancy was maintained to expose the model to a broad spectrum of protein contexts and ligand environments. After filtering for steric clashes, 293,019 high-confidence complexes were retained.
- FDA-AD dataset: Unique crystal structures were defined as unique FDA drug–protein pairs. Redundant structures for the same drug–target pair were removed, but multiple targets of the same drug are retained.

- DEL dataset: A one-to-one mapping was applied by retaining only the highest-affinity DEL hit per target, resulting in 128 unique DEL–target structures.

These clarifications have been added to Table S1 and incorporated into Section 4.1 of the revised manuscript, which now reads as follows:

“To ensure methodological consistency, we note that redundancy was handled differently across datasets depending on their intended use. Specifically, the BioLiP2 and AlphaFill datasets intentionally retain multiple structures bound to the same ligand, thereby exposing the model to diverse conformational states and binding environments. In contrast, for the FDA-AD and DEL datasets we applied stricter curation: the FDA-AD set was reduced to unique FDA drug–protein pairs (removing redundant crystal structures for the same drug–target), while the DEL set was limited to one highest-affinity hit per target protein.”

9. The Reviewer commented:

Comments:

Page 2, line 15 states that “few DEL-derived compounds have advanced to clinical trials or reached the market.” I disagree with that assertion. I would refer the authors to a nice review written by David Liu in which he performed a comprehensive analysis of DEL derived molecules that have entered the clinic. Nat Rev Drug Discov. 2023 Jun 16;22(9):699–722. doi:10.1038/s41573-023-00713-6. While the total number of DEL molecules that have reached the clinic is comparatively low relative to molecules identified by high-throughput screening, DEL has only recently been adopted as a standard approach for hit identification.

Author reply:

We thank the reviewer for this helpful comment and for referring us to the review by David Liu et al. (Nat Rev Drug Discov. 2023, cited as Ref. 7). We agree that our original phrasing did not fully capture the clinical progress achieved with DEL-derived compounds. To address this, we have revised the abstract and introduction to more accurately reflect the current status while noting that DEL is a relatively recent approach for hit identification:

“DNA-encoded libraries (DELs) facilitate high-throughput screening of trillions of molecules against protein targets through split-pool synthesis and DNA tagging. Despite their potential, only a few DEL-derived compounds have advanced to clinical trials or reached the market. A better understanding of the defining characteristics of target proteins, particularly those with binding pockets suitable for DEL screening, is critical to improving success rates.” (Abstract)

“Despite the high-throughput capabilities and economic advantages of the DEL technique, the number of DEL-derived molecules advancing to clinical trials or reaching the market remains relatively low, partly due to the relatively recent adoption of DEL as a mainstream approach for hit identification, as well as persistent challenges such as uncertainties in processing DEL data, difficulties in optimizing DEL hits, and a general lack of understanding on target druggability with DEL molecules⁷.” (Introduction)

10. The Reviewer commented:

The authors included structures from three databases of experimentally derived structures, BioLiP2, FDA and OpenDEL, and two databases of predicted structures, AlphaFill and AlphaFold. It would be interesting to know why they didn't include in their comparisons the human protein structures from the Protein Data Bank with small molecule ligands not included in the BioLiP2, FDA and OpenDEL databases.

Author reply :

We thank the reviewer for this comment. While the PDB contains many protein–ligand complexes, a substantial fraction of ligands are either crystallization artifacts or lack biological relevance. BioLiP2 is a curated subset of the PDB that retains only biologically meaningful interactions with functional annotation. To ensure reliability and avoid noise, we therefore based our analyses on BioLiP2, along with the FDA-AD and the DEL datasets we prepared. We have clarified this point in Methods Section 4.1 as follows:

“We specifically chose BioLiP2 rather than using all PDB entries with ligands because BioLiP2 systematically filters out crystallization artifacts and non-functional ligands, thereby ensuring greater biological relevance and reliability.”

11. The Reviewer commented:

Section 2.5 described the identification of pockets from the human proteome that are suitable for DEL screening. The set of AlphaFold2 predicted human protein structures was filtered prior to analysis with ErePOC2. One of the filters applied was to remove proteins that had pocket volumes less than 800 Å³. On page 7, line 151 they indicate that the average volume of a DEL molecule binding pocket is 2723.3 Å³. They should explain why they decided to use 800 Å³ as their volume cutoff.

Author reply :

We thank the reviewer for this helpful comment. Previous studies have suggested that a minimum pocket volume of ~500 Å³ is required to accommodate drug-like small molecules (e.g., J. Phys. Chem. B 2022, 126, 6853–6867). We adopted a more stringent threshold of 800 Å³ to exclude very small, shallow, or solvent-exposed cavities, thereby retaining pockets most likely to be DEL-compatible. This is motivated by the observation that DEL-binding pockets are, on average, about 1.3 times larger than FDA-AD pockets.

Among 862,347 pockets initially detected in AlphaFold2-predicted human proteins, this filter removed ~37% of the smallest cavities. We further filtered pockets with average pLDDT < 0.7, resulting in 182,424 high-confidence pockets for ErePOC analysis. We have added one additional sentence in the main text (Section 2.5) accordingly to reflect this rationale.

“The 800 Å³ threshold was selected based on prior studies suggesting 500 Å³ as a minimal druggable pocket volume³², together with our observation that DEL-binding pockets are

substantially larger.”

12. The Reviewer commented:

Figs. 2A and S10A are lacking units for the X-axis.

Figs. 2C and S10C the legend below the graph has a typo: “polor” should be “polar”

Fig. S3 should include a legend indicating the corresponding binding sites and colors as was done in Fig. 4.

Author reply :

We thank the reviewer for her or his careful reading and for identifying these omissions and errors. We have now updated the figures as follows: units have been added to the X-axes in Figs. 2A and S13A (the original S10A); the typo in Figs. 2C and S13C (the original S10C) has been corrected to "polar"; and a detailed legend clarifying the binding sites and color scheme has been added to Fig. S5 (the original S3), consistent with the style of Fig. 4. These revisions have been incorporated into the revised manuscript.

Response to Reviewer #2

1. The Reviewer commented:

The authors describe a method to represent the properties of protein pockets that captures both structural and functional features, and use this method to investigate DEL target space.

The paper is well written, but we have a number of comments around the methodology.

Author reply:

We sincerely thank the reviewer for carefully reading our manuscript and for providing thoughtful comments. We appreciate the constructive feedback, which has helped us improve the clarity and rigor of our work.

2. The Reviewer commented:

Comments:

The DEL dataset contains 128 entries. It would be useful if the authors included the associated SMILES strings for the ligands and a depiction of their structures (as supplementary information).

Author reply:

We thank the reviewer for this helpful suggestion. The SMILES strings of the 128 DEL ligands have been added as Supplementary Table S2, and the Methods section (Section 4.1) has been updated accordingly:

“This dataset ultimately consisted of 128 unique DEL hit–target protein structures. The corresponding SMILES strings of these ligands are provided in Supplementary Table S2.”

3. The Reviewer commented:

“We hypothesize that DEL targets exhibit unique pocket characteristics that can be utilized in drug design decision-making, such as target prioritization and hit identification.” Does this study shed any light on these unique pocket characteristics? Or indeed is the conclusion that the pocket characteristics are similar to general druggable pocket?

Author Reply:

We thank the reviewer for this insightful question. Our results reveal that DEL pockets indeed differ statistically from general druggable pockets, although these differences are subtle and partially overlapping rather than categorically distinct. This observation highlights the need for developing a continuous representation such as ErePOC, which provides a more informative characterization of pockets than conventional geometric or physicochemical feature descriptors.

Specifically, we conducted additional quantitative analyses of pocket features (volume, number of alpha spheres, alpha sphere density, apolar alpha sphere proportion, mean local hydrophobic density, proportion of polar atoms, and druggability) and pocket–ligand interaction features (hydrophobic, hydrogen-bond, and polar contacts) across DEL, FDA-approved, and BioLiP2

pockets. Statistical analyses using Cliff's δ effect sizes and p-values, followed by principal component analysis (PCA), were performed to quantify and visualize their differences.

Our results show that DEL pockets are generally larger and structurally more complex, with a hydrophobic-dominated binding environment complemented by selective polar contacts. Statistical comparisons (Figure S3A) confirm that DEL pockets are significantly larger than FDA-AD targets (volume $\delta = 0.405$; number of alpha spheres $\delta = 0.409$; alpha sphere density $\delta = 0.395$; $p < 3.6e-11$) and BioLiP2 pockets (volume $\delta = 0.302$; number of alpha spheres $\delta = 0.321$; alpha sphere density $\delta = 0.201$; $p < 3.4e-09$), indicating that DEL screening identifies molecules interacting with larger and more complex binding interfaces. DEL pockets also exhibit a balanced polar–apolar composition: the proportion of apolar alpha spheres is slightly lower than in FDA-AD targets ($\delta = -0.155$) but higher than in BioLiP2 ($\delta = 0.193$). Mean local hydrophobic density ($\delta = 0.136$ vs FDA-AD, $\delta = 0.342$ vs BioLiP2), proportion of polar atoms ($\delta = 0.177$ vs FDA-AD, $\delta = -0.137$ vs BioLiP2), and druggability score ($\delta = -0.109$ vs FDA-AD, $\delta = 0.045$ vs BioLiP2) further highlight their intermediate physicochemical profile. Analysis of pocket–ligand interactions (Figure S3B) demonstrates that DEL pockets are dominated by hydrophobic contacts ($\delta = 0.122$ vs FDA-AD, $\delta = 0.378$ vs BioLiP2), whereas hydrogen bonds ($\delta = -0.150$ vs FDA-AD, $\delta = -0.392$ vs BioLiP2) and polar interactions ($\delta = -0.207$ vs FDA-AD, $\delta = -0.459$ vs BioLiP2) are reduced. These characteristics indicate that hydrophobic effects drive DEL binding, stabilized by a minimal yet functionally critical set of polar anchors.

PCA analysis further shows that DEL pockets cluster in a distinct but partially overlapping region of chemical space. These updated results are summarized in Section 2.2 as follows, with supporting data provided in two new Supplementary Figures (S3 and S4).

“Statistical analyses were performed by computing Cliff's δ effect sizes for key pocket and pocket-ligand interaction features (Figure S3). These analyses confirmed that DEL pockets were significantly larger than FDA-AD targets (volume $\delta = 0.405$; number of alpha spheres $\delta = 0.409$; alpha sphere density $\delta = 0.395$; $p < 3.6e-11$) and BioLiP2 pockets (volume $\delta = 0.302$; number of alpha spheres $\delta = 0.321$; alpha sphere density $\delta = 0.201$; $p < 3.4e-09$), and exhibited a balanced polar–apolar composition, with physicochemical properties intermediate between FDA-AD and BioLiP2 (Figure S3A). Analysis of pocket–ligand interactions confirmed that DEL pockets were dominated by hydrophobic contacts ($\delta = 0.122$ vs FDA-AD, $\delta = 0.378$ vs BioLiP2), whereas hydrogen bonds ($\delta = -0.150$ vs FDA-AD, $\delta = -0.392$ vs BioLiP2) and polar interactions ($\delta = -0.207$ vs FDA-AD, $\delta = -0.459$ vs BioLiP2) were reduced (Figure S3B). These characteristics indicate that DEL binding is primarily driven by hydrophobic effects, stabilized by a minimal yet functionally critical set of polar anchors.

Principal component analysis (PCA) of features further confirmed these patterns, showing that DEL pockets occupy a distinct region in PCA space (Figure S4). PC1 primarily reflected chemical composition, including apolar/polar atom ratios and interaction types, while PC2 was dominated by structural size descriptors, together accounting for approximately 75% of the variance. The pocket analysis was also consistent with molecular property differences

obtained using MOE²⁹ (Figure S2). DEL molecules exhibited lower water solubility ($\text{LogS} = -6.49$) compared to approved drugs (-3.05) and higher hydrophobicity ($\text{cLogP} = 3.42$ vs. 1.44). While DEL pockets shared the overall druggability characteristics of FDA-AD targets, they displayed a distinct physicochemical bias, favoring larger, more hydrophobic pockets with a different polar–apolar balance. We note that no single feature or simple combination could distinguish DEL from FDA-AD or general protein pockets, possibly due to the broad variability of pocket structures. This observation highlights the need for developing a more informative representation of pockets.” (Section 2.1)

Figure S3. Statistical comparison of pocket and interaction features. **A.** Pocket features (volume, alpha spheres, density, polar/apolar composition, hydrophobicity, druggability) compared between DEL, FDA-AD, and BioLiP2 targets using Cliff’s delta; significance: * $p < 0.05$, ** $p < 0.01$, *** $p < 0.001$; **B.** Interaction features (hydrophobic, hydrogen-bond, and polar contacts) compared across the three datasets using Cliff’s delta with the same significance notation.

Figure S4. Principal component analysis (PCA) of pocket and interaction features. DEL (blue) and FDA-AD (red) targets were projected onto the PCA space trained using BioLiP2 pockets (gray background). Kernel density contours highlight the distinct yet partially overlapping distributions of DEL and FDA-AD targets. PC1 is primarily influenced by chemical composition, including apolar–polar balance and interaction types (e.g., apolar α -sphere proportion 4.1%, proportion of polar atoms 4.0%, hydrophobic 3.8%, hydrogen bonds 3.9%, polar interactions 4.0%), whereas PC2 is dominated by structural size descriptors (pocket volume 11.7% and number of alpha spheres 11.2%). Together, these components capture the separation of DEL targets from reference datasets.

4. The Reviewer commented:

P6, L123: “DEL and FDA-AD pockets were notably larger than those in BioLiP2 and AlphaFill, likely due to differences in ligand characteristics.” I do not believe that this shows that the pockets are larger, merely that the ligands are larger, and hence the number of residues within 5Å. The pocket size is accurately assessed in Section 2.2.

Author reply:

We thank the reviewer for pointing this out. We agree that the number of residues primarily reflects ligand size rather than true pocket volume, which is more accurately assessed in the subsequent

Section 2.2. Accordingly, we have revised the sentences as follows:

“As shown in Fig. 1A, the average number of residues in BioLiP2, AlphaFill, DEL, and FDA-AD pockets was 12.5, 12.5, 28.1, and 16.1, respectively. The larger number of residues surrounding DEL and FDA-AD ligands likely reflects their greater molecular size and chemical complexity, often containing halogen atoms and other bulky functional groups that demand more spatially extended binding environments. The average molecular weights of DEL and FDA-AD ligands in this study are 560.5 and 310.9 Da, respectively (Figure S2). In contrast, regular ligands and their pockets have co-evolved for an optimized fit tuned to biological needs rather than maximal binding.”

5. The Reviewer commented:

P7: Across the various properties, is the difference between DEL and FDA-AD statistically significant? Is there any evidence that the targets for which DEL has been run differ from the set of all druggable targets?

Author reply:

We thank the reviewer for raising these important questions. As detailed in our response to Comment #3, we performed statistical analyses comparing DEL and FDA-AD pockets, confirming that DEL pockets are significantly larger and more hydrophobic than FDA-AD pockets (e.g., Volume $\delta = 0.405$, $p < 3.6e-11$). PCA further shows that DEL targets occupy a distinct region within the broader druggable pocket space.

Regarding the second question, there is currently no reliable evidence that targets screened by DEL differ systematically from the broader set of druggable proteins, although it is commonly believed that membrane proteins are less frequently screened with DEL in contrast with enzymes. In practice, it is unknown how many targets — and which ones — have already been subjected to DEL screening, as such information is rarely disclosed unless a hit is identified, further optimized, and ultimately published. Negative results are generally not reported, and large-scale DEL screening efforts in industry are typically proprietary. As a result, publicly available datasets are inherently biased toward successful cases.

6. The Reviewer commented:

The FDA-AD compounds are the result of a lead optimisation process that would have eliminated functional groups that are problematic for metabolism. The DEL compounds are investigational compounds and their structures are expected to be less optimal. This might explain why ionic interactions were found in the DEL dataset. If the authors had included the associated ligand structures for the DEL dataset, this would be easier to assess.

Author reply:

We thank the reviewer for this insightful suggestion. We have now included the ligand SMILES for the DEL dataset in the SI (Table S2), which will allow easier assessment of the chemical and interaction features. We agree that the higher proportion of ionic interactions observed in the DEL dataset compared to the FDA-AD set likely reflects the fact that DEL hits have not

undergone further optimization for metabolic stability, whereas FDA-approved drugs have. We also note that the number of ionic interactions in the FDA-AD dataset changed slightly after applying a more stringent similarity criterion to define FDA-approved drugs, as suggested by the reviewer. Accordingly, we have revised the sentence in the manuscript to:

“Ionic interactions were slightly more prevalent in DEL pockets (1.3%) than in FDA-AD pockets (0.7%), while BioLiP2 pockets exhibited a higher fraction of ionic interactions (3.9%). This pattern may reflect the early, pre-optimization status of DEL compounds compared to approved drugs” (Section 2.2)

7. The Reviewer commented:

The phrase “DEL pockets” seems a bit misleading. These are just pockets where DEL ligands have been reported. Presumably, small molecules have also been reported for these pockets, or is this not the case?

Author reply:

We thank the reviewer for this insightful comment. Yes, as correctly noted, the term “DEL pockets” refers to pockets where DEL-derived ligands have been reported, not to pockets that exclusively bind DEL ligands. In many cases, these pockets also accommodate other small molecules or metabolites such as ATP. We have clarified this in the revised manuscript as follows.

To avoid ambiguity, we have revised Section 4.1 of the Methods to clarify this terminology:

“It should be noted that in this study, “DEL pockets” refer to binding sites identified through reported DEL-derived ligands, which may nonetheless also bind other small molecules or endogenous metabolites.”

8. The Reviewer commented:

P 7, L174: “DEL pockets possess unique features in their physicochemical properties, distinguishing them from general protein pockets.” The results shown seem to indicate that their properties are similar to those for FDA-AD.

Author reply:

We thank the reviewer for this valuable comment. As noted in our earlier responses, DEL and FDA-AD pockets do share the common property of druggability, as reflected by their partial overlap in PCA space. However, statistical analyses reveal that DEL pockets are significantly larger and exhibit a bias toward hydrophobic interactions, with fewer polar contacts. As this part of the manuscript has been revised, the sentence has been replaced by the following text (Section 2.2):

“While DEL pockets shared the overall druggability characteristics of FDA-AD targets, they displayed a distinct physicochemical bias, favoring larger, more hydrophobic pockets with a different polar–apolar balance. We note that no single feature or simple combination could

distinguish DEL from FDA-AD or general protein pockets, possibly due to the broad variability of pocket structures. This observation highlights the need for developing a more informative representation of pockets.”

9. The Reviewer commented:

Fig 5: If prediction of human protein targets suitable for DEL screening had been done instead with the Drug-AD pockets would there have been any difference in the results? If so, this would be interesting to report. What I am trying to figure out is whether the ‘DEL-able’ pocket space really is different than general druggable pocket space. On P11, it is stated that “Together, these results suggest that DEL screening can readily access most of the known druggable target space.” which would indicate that there is little difference.

Author reply:

We thank the reviewer for raising this important and thought-provoking comment. We have performed the suggested control analysis by repeating our prediction pipeline using FDA-AD reference pockets in place of DEL reference pockets. Consistent with expectations, both FDA-AD and DEL-based predictions identified a largely overlapping set of “druggable” human protein pockets. However, important differences emerged, as shown by the enrichment of protein families containing pockets with high similarity (cosine > 0.8) to either FDA-AD or DEL reference pockets (Figure S18). While enzyme classes such as **transferases, hydrolases, and oxidoreductases** were enriched in both, we identified divergent preferences in other protein families. In particular, FDA-AD-like pockets clustered in classical targets (**receptors, ion channels, and isomerases**), whereas DEL-like pockets were uniquely enriched in families such as **RNA-binding proteins, chromatin regulators, and GTPase activators**.

We also include an illustrative case comparing MAT2A and MAT2B within the methionine adenosyltransferase family. MAT2A—despite being a validated drug target—harbors a small, polar pocket with relatively low similarity to DEL reference pockets (maximum cosine similarity = 0.66). In contrast, MAT2B contains a larger, more hydrophobic pocket with much higher DEL similarity (cosine = 0.93; Figure S19). Following Reviewer 1’s suggestion, we also performed *in silico* DEL screening by docking a pre-assembled 2.8-million DEL compound library against both MAT2A and MAT2B. Consistent with ErePOC predictions, MAT2B yielded a substantially more favorable mean docking score (−8.8 vs −5.3 kcal/mol), indicating a higher likelihood of DEL compatibility.

In summary, DEL-compatible pockets largely overlap with the general druggable space but exhibit a distinct bias toward larger and more hydrophobic cavities. This complementary perspective underscores how DEL can extend beyond traditional drug discovery paradigms. We have added the following revised text to the main manuscript:

Section 2.4

“Together, these results suggest that DEL screening can access the majority of known druggable targets, while exhibiting a measurable preference for pockets with features conducive to DEL hit generation, such as larger volume and higher hydrophobicity.” (Section

2.4)

Section 2.5

“The FDA-AD dataset, which encompasses 31 protein families, showed a similarly broad distribution, with transferases (20.8%), hydrolases (18.1%), oxidoreductases (14.8%), DNA-binding proteins (7.3%), and receptors (6.9%) as the most frequent classes. ... Interestingly, transferases, hydrolases, and oxidoreductases are consistently enriched across all three datasets (DEL, FDA-AD, and predicted DEL-compatible pockets), underscoring their central roles as preferred families for DEL screening.” (Section 2.5)

Conclusion and Discussion

“To clarify how DEL-compatible pockets relate to the traditional druggable space, we repeated our ErePOC cosine similarity analysis using FDA-AD reference pockets in place of DEL-derived ones. While both sets identified overlapping human pockets, enrichment patterns revealed distinct preferences. Among predicted high-similarity pockets, FDA-AD-like pockets were predominantly enriched in classical target families such as receptors, ion channels, and isomerases, whereas DEL-like pockets were uniquely enriched in families including RNA-binding proteins, chromatin regulators, and GTPase activators (Figure S18). Notably, transferases, hydrolases, and oxidoreductases were enriched by both predictions, highlighting shared accessible space. The relative scarcity of DEL-compatible receptor and channel targets may reflect the practical challenges of screening membrane proteins in DEL experiments. These results suggest that although DEL screening overlaps with general druggable space, it exhibits a measurable bias toward target classes with larger, more hydrophobic and chemically tractable binding pockets. For example, within the methionine adenosyltransferase family, the regulatory MAT2B harbors a larger, more hydrophobic pocket than the catalytic MAT2A (Figure S19). Despite MAT2A being a validated drug target³⁹, its pocket showed lower similarity to DEL reference pockets (cosine = 0.66) compared to MAT2B (cosine = 0.93), suggesting it may be less suitable for DEL screening. In silico DEL screening using docking confirmed that MAT2B exhibited a substantially more favorable mean docking score (−8.8 vs −5.3 kcal/mol). Together, these results support the ability of the ErePOC framework to identify targets with better DEL compatibility, enabling more efficient and strategically focused applications of DEL in early-stage drug discovery.”

Figure S18. Comparative analysis of human protein enrichment using pocket similarity with either DEL or FDA-AD pockets using ErePOC representation (cosine similarity > 0.8)

A. MAT2A
Cosine similarity with DEL
pocket (AASS): 0.66

Volume: 927 Å³
Hydrophobicity score: -4.9

B. MAT2B
Cosine similarity with
DEL pocket (IRS): 0.93

Volume: 4672 Å³
Hydrophobicity score: 21.9

Figure S19. Comparative Analysis of Pocket Properties between MAT2A and MAT2B. A. The structure and pocket of MAT2A; B. The structure and pocket of MAT2B; C. Distribution of docking scores; D. Distribution of docking Z-scores for MAT2A (blue) and MAT2B (red).

10. The Reviewer commented:

P. 13: “These examples highlight how contrastive learning imparts the capacity to detect higher-level functional or physicochemical similarities that are not solely reflected by either global or local 3D structural alignment.” Are these not false positives, or is there experimental evidence that these bind very similar ligands?

Author reply:

We thank the reviewer for this valuable comment. We agree that, without additional experimental validation, some examples shown in Figure 6 could present false positives. These cases do not directly demonstrate functional equivalence or ligand binding similarity per se, but

rather illustrate how embedding-based similarity derived from contrastive learning may capture higher-level functional or physicochemical relationships that are not apparent from global or local 3D structural alignment alone.

This interpretation aligns with prior findings that pockets accommodating the same ligand (e.g., ATP) can exhibit substantial geometric variation (Kahraman et al., 2007) and that functional convergence can occur across distinct structural folds (Söding et al., 2008). To reflect this nuance and avoid overinterpretation, we have revised the relevant paragraph in the main text (Section 2.6) as follows:

“These cases indicate that contrastive learning may capture potential functional or physicochemical relationships between binding pockets that are not fully explained by global protein fold or local geometric similarity. Comparable observations have been reported in earlier studies, where pockets binding the same ligand (e.g., ATP) exhibit considerable geometric diversity²⁰, and functional associations are detectable across distinct structural folds³⁵. Although further experimental evidence will be required to substantiate such relationships in our predictions, these findings suggest that embedding-based similarity could provide information complementary to conventional structure alignment methods and may offer new hypotheses for future exploration.”

11. The Reviewer commented:

P17: “A ligand was considered an FDA-approved drug if its Tanimoto similarity exceeded 0.7.” Unless the similarity is 1.0, the structure is different. It may be a tautomer, in which case you could use the InChI to check for identity. But otherwise, these will not be the same molecule, and I believe that it is incorrect to state that these are FDA-approved drugs. They simply have high similarity to FDA-approved drugs.

Author reply:

We thank the reviewer for highlighting this important point. We agree that a Tanimoto similarity threshold < 1.0 could include non-identical molecules, such as tautomers or close analogs. To avoid this ambiguity, we have revised our workflow to require a Tanimoto similarity of exactly 1.0, ensuring chemical identity between PDB ligands and FDA-approved drugs. This stricter criterion guarantees that only structurally identical molecules are retained, excluding compounds with even minor differences. As a result, the updated FDA-AD dataset now contains 340 (instead of the original 630) unique crystal structures of drug–pocket complexes, each representing a distinct FDA-approved drug–protein pair. All related statistics in the manuscript have been updated accordingly. This revision is reflected in the revised Methods section as follows:

“Our third dataset focused on FDA-approved drugs, collected from the DrugBank database. We gathered information on 2,863 FDA-approved drugs and used their SMILES representations to search the Protein Data Bank (PDB) for corresponding drug–protein complexes. In total, 5,286 potential drug–protein complex structures were identified. Ligands in these structures were marked using the ‘HET’ keyword, and Tanimoto similarity based on Morgan fingerprints

(radius = 2) was calculated using RDKit between these ligands and the FDA drug list. A ligand was considered an FDA-approved drug only if its Tanimoto similarity equaled 1.0, ensuring exact chemical identity. Pockets were defined as amino acid residues within 5 Å of these ligands, resulting in a final dataset of 340 unique crystal structures, each corresponding to a distinct FDA-approved (FDA-AD) drug–protein pair. Multiple targets of the same drug were retained, but redundant crystal structures of the same drug–target combination were removed.” (Page 40, Section 4.1)

12. The Reviewer commented:

P5, line 112: "i) the 112 BioLiP2(22) database, which provides experimentally determined binding structures of natural ligands and their associated pockets."

This sentence is confusing, as BioLip2 is not limited to natural ligands. If authors used a subset of BioLip2, it should be clearly stated here. Method section, line 438 mentions the filter for “regular ligands”, but this is not clear in the previous sections.

Author Reply:

We thank the reviewer for pointing out this ambiguity. Indeed, BioLiP2 includes a broad range of ligand types, not just natural ligands. In our study, we specifically restricted the dataset to “regular ligands” as defined in the BioLiP2 annotation, excluding crystallization additives, metal ions, and other non-functional or non-biologically relevant molecules. To clarify this, we have revised the text in Section 2.1 to explicitly state that we used the subset of BioLiP2 entries with regular ligands. The revised sentence now reads:

“(i) the BioLiP2²³ database, from which we selected entries annotated with regular ligands (i.e., biologically relevant small molecules) and their associated pockets;”

We have also updated the manuscript throughout to replace instances of “natural ligands” with “regular ligands” to more accurately reflect the dataset used.

13. The Reviewer commented:

*P5, line 118: "They include 326,416, 330,434, 128, and 630 pockets, respectively"
The study's comparative analysis is challenged by the data imbalance, with the BioLip2 and AlphaFill datasets being orders of magnitude larger than the primary DEL dataset (n=128).
Could the authors comment on this and whether it may effect the statistical power of the conclusion?*

Author reply:

We thank the reviewer for raising this important point regarding dataset size imbalance. Indeed, the BioLiP2 and AlphaFill datasets are substantially larger than the DEL dataset (n = 128). To ensure the robustness of our comparative analyses despite this imbalance, we employed statistical methods that remain valid under unequal sample sizes. Specifically, we used non-parametric tests such as the Mann–Whitney U test for comparing medians, and permutation tests to assess distributional differences (Figure S3). These approaches rely on the underlying

data distributions and are less sensitive to sample size, making them appropriate for our study.

Moreover, key physicochemical properties—such as pocket volume, hydrophobicity, and polarity—showed consistently significant differences ($p < 0.05$, often $p < 0.001$) across all comparisons. This indicates that the observed trends are robust and unlikely to be artifacts of data size imbalance. Essentially, while the DEL dataset is relatively small, it is not prohibitively so ($n=128$), and the magnitude and consistency of its distinctive physicochemical features enabled statistically significant separation from larger background datasets, supporting the validity of our conclusions.

14. The Reviewer commented:

P5, 118-123: For the BioLiP2 dataset, pockets are defined functionally based on "experimentally annotated binding residues," which typically includes a small set of specific interacting residues. For all other datasets, pockets are defined geometrically using a universal 5 Å distance cutoff from the ligand which creates a more general, inclusive cavity. This inconsistency introduces a systematic bias in the average number of residues observed in Figure 1A.

Author reply:

We sincerely thank the reviewer for raising this important concern. In our study, binding pockets in the FDA-AD, DEL, and AlphaFill datasets were consistently defined using a 5 Å distance cutoff from ligand atoms. The BioLiP2 dataset is annotated with binding pockets, so we originally used the curated functional annotations provided by the database. To ensure methodological consistency and assess robustness, we additionally redefined BioLiP2 pockets using the same 5 Å geometric criterion applied to the other datasets. This unified definition was used to re-analyze all datasets and consistent results were obtained between these two BioLiP2 pocket definition.

In the revised manuscript (Results, Section 2.1), we now clarify both definition:

"For BioLiP2, pockets were defined using experimentally annotated binding residues provided by the webserver. Meanwhile, pockets for the AlphaFill, DEL, and FDA-AD datasets were generated by including all amino acid residues within 5 Å of the bound ligand. To assess the impact of this inconsistency, we additionally redefined BioLiP2 pockets using the same distance-based criterion and repeated all analyses under this unified definition. The key findings remained consistent across definitions, suggesting that our conclusions are reasonably robust to differences in pocket definition."

We have also added the following details in the Methods section (4.1):

"In BioLiP2, binding sites are originally defined based on functional residue annotations. Specifically, a protein residue is labeled as a binding residue if at least one of its atoms is within a distance equal to the sum of the van der Waals radii plus 0.5 Å of any atom of the bound ligand, and if there are at least two such inter-atomic contacts, the residue is considered part of the binding site. To ensure methodological consistency across datasets, we additionally defined binding pockets using a simple geometric criterion, including all protein atoms within

5 Å of any non-hydrogen atom of the bound ligand. All analyses were repeated using this definition, and the results remained consistent across both definitions (Figs. S8–S10 and S14–S15).”

Under this unified pocket definition, the average number of residues per pocket was recalculated: BioLip2 increased from 9.1 to 12.5 residues, matching AlphaFill (12.5). The DEL (28.1) and FDA-AD (16.1) pockets still remain larger than BioLip2 and AlphaFill, reflecting the generally larger and more chemically complex nature of synthetic drug molecules and DEL hits compared to regular ligands. While minor quantitative changes were observed (e.g., slightly lower enrichment of charged residues and slightly higher hydrophobic residue proportions in BioLip2), the overall trends and conclusions remain unchanged: DEL pockets are larger and more hydrophobic, exhibiting weaker polar and hydrogen-bond interactions than BioLip2 pockets. We have also tested the robustness of ErePOC against these two different pocket definitions, and found very consistent results (Figure S8-S10, S14-S15). Accordingly, we have added the following sentences to the Section 2.3:

“We note that for ErePOC model training and downstream analyses, the original BioLip2 binding residue annotations were used to preserve experimentally validated contacts. Training and analyses using the alternative 5 Å pocket definition yielded highly similar results, as demonstrated in Figure S8-S10.”

15. The Reviewer commented:

P7, line 164: “We further analysed the interactions between pocket residues and ligands using the Arpeggio method(25) (Figure 2 a-c). Hydrophobic interactions accounted for 50.7% in DEL, 49.5% in FDA-AD, and 30.1% in BioLip2. Hydrogen bond interactions were 3.8% for DEL, 4.8% for FDA-AD, and 9.9% for BioLip2, while polar interactions constituted 6.0%, 7.8%, and 14.8%, respectively. Ionic interactions can be found in the DEL dataset (1.50%) but were almost absent in FDA-AD (0.15%).”

The authors report "Hydrogen bond interactions" (e.g., 9.9% in BioLip2) and "polar interactions" (14.8% in BioLip2) as two distinct categories. This is confusing because a hydrogen bond is a specific and strong type of polar interaction. What does the "polar interactions" category actually contain if it excludes hydrogen bonds?

Author reply:

We thank the reviewer for this insightful comment. In Arpeggio, hydrogen bonds are defined as strong, directional polar interactions that satisfy specific distance and angle criteria. The “polar interactions” category, by contrast, encompasses contacts between polar atoms that do not meet these stringent criteria and thus are not classified as hydrogen bonds. In addition, “weak polar interactions” represent even more relaxed polar contacts. While all three categories involve polar atoms, Arpeggio distinguishes them to enable a more nuanced characterization of ligand–pocket interactions. We have now clarified this distinction in the manuscript as follows:

“The interaction types included in our analysis were van der Waals, hydrogen bonds, weak hydrogen bonds, halogen bonds, ionic interactions, aromatic interactions, hydrophobic interactions, carbonyl interactions, polar interactions (contacts between polar atoms that do not meet the geometric criteria for hydrogen bonds), and weak polar interactions.” (Page 22)

16. The Reviewer commented:

Moreover, the authors state that ionic interactions are more common in the DEL set (1.50%) than the FDA-AD set (0.15%). Given the tiny size of the DEL dataset (n=128), could this 1.5% be an artifact arising from just a few charged ligands in their small sample? Also, the corresponding value for the massive BioLiP2 dataset should be reported?

Author reply:

We thank the reviewers for these valuable comments and for prompting a closer examination of this point. First, we refined the FDA-AD dataset to include only exact matches to FDA-approved drugs (Tanimoto similarity = 1), reducing the dataset from 630 to 340 targets. In this filtered dataset, 24 FDA-AD targets exhibited ionic interactions, corresponding to 102 atomic pairs classified as ionic by Arpeggio. These account for 0.7% of all ligand–protein atomic interaction pairs (102 out of 13,862). In comparison, 34 targets in the DEL dataset contained ionic interactions, totaling 116 atomic pairs, or 1.3% of all ligand–protein interactions (116 out of 8,903). For BioLiP2, ionic interactions comprised 3.9% of the total.

These results confirm that ionic interactions are consistently observed across all datasets and are unlikely to be artifacts of sample size. The slightly higher frequency in the DEL dataset likely reflects the investigational, pre-optimization status of these compounds, which encompass broader chemical diversity and have not yet been optimized for metabolic stability. The revised manuscript now incorporates this clarification:

“Ionic interactions were slightly more prevalent in DEL pockets (1.3%) than in FDA-AD pockets (0.7%), while BioLiP2 pockets exhibited a higher fraction of ionic interactions (3.9%). This pattern may reflect the early, pre-optimization status of DEL compounds compared to approved drugs.” (Section 2.2)

17. The Reviewer commented:

P16, line 446: The low-quality dataset from AlphaFill is a critical issue that is not adequately addressed. The authors' only acknowledgement of this problem is a single sentence stating that the "quality and resolution of the predicted complexes were relatively low" due to "potential clashes." It appears that the authors used all 330,434 structures without any quality-based filtering. Surely this brings into question the results from the AlphaFill analysis as the defined pockets may be artifacts or the wrong size.

Author reply:

We thank the reviewer for raising this important point. We fully agree that the structural quality of AlphaFill entries is relatively low compared to the other three datasets. This is precisely why most of our analyses are built upon on BioLiP2, DEL and FDA-AD datasets, with AlphaFill

results serving only as a secondary comparison.

Following the reviewer's suggestion, we have now implemented a quality-control procedure based on steric clash detection using an in-house Python script built on Biopython's Bio.PDB. Structures were discarded if the pocket clash score was ≥ 0.005 , or the ligand clash score ≥ 0.01 , or the number of protein–ligand clashes ≥ 5 , resulting in the removal of $\sim 11\%$ of entries and yielding 293,019 relatively high-quality complexes (Methods, Section 4.1). We repeated all relevant analyses on this filtered dataset and observed only minor changes. For example, the average number of residues per pocket changed slightly from 12.88 to 12.46. All relevant statistics have been updated in the manuscript, along with Figures 1, S6, and S10.

Corresponding revisions have been made to the Methods (Section 4.1) as follows:

“Additionally, we curated predicted ligand-protein complex structures from the AlphaFill database. AlphaFill transfers ligands, cofactors, and metal ions to AlphaFold-predicted models based on sequence and structural similarities. However, due to potential clashes between the transplanted ligands and the predicted receptor structures, the quality and resolution of the predicted complexes were relatively low. To improve the structural quality of AlphaFill entries, we implemented a steric clash detection procedure using an in-house Python script. Steric clashes were defined as any pair of non-hydrogen atoms separated by less than 1.2 Å. For each pocket and ligand, we calculated a clash score as the fraction of atom pairs that fell below this threshold. Protein–ligand clashes were quantified as the number of atom pairs between the pocket and ligand with interatomic distances shorter than 1.2 Å. Structures were discarded if the pocket clash score was ≥ 0.005 , or the ligand clash score was ≥ 0.01 , or if the number of protein-ligand clashes was ≥ 5 . This procedure removed approximately 11% of the AlphaFill dataset, yielding 293,019 relatively high-quality ligand–protein complexes for downstream analysis.”

18. The Reviewer commented:

P16, line 454-457: “Ligands in these structures were marked using the 'HET' keyword, and Tanimoto similarity based on molecular fingerprints was calculated 455 between these ligands and the FDA drug list. A ligand was considered an FDA-approved drug if its 456 Tanimoto similarity exceeded 0.7.”

What fingerprint (FP) was used when doing the Tanimoto calculation? Without knowing this, the score thresholds are meaningless as interpretation varies from one FP to another. In any case, 0.7 seems very low, and it may find similar compounds but not the exact molecule.

Author reply:

We thank the reviewer for highlighting this important point. As noted in our response to Comment #11, we now set the threshold to 1.0, ensuring exact chemical identity. The Tanimoto similarity was computed using RDKit with Morgan fingerprints (radius = 2), which is now explicitly stated in the Methods section (Section 4.1):

“Ligands in these structures were marked using the ‘HET’ keyword, and Tanimoto similarity based on Morgan fingerprints (radius = 2) was calculated using RDKit between these ligands and the FDA drug list. A ligand was considered an FDA-approved drug only if its Tanimoto similarity equaled 1.0, ensuring exact chemical identity.”

19. The Reviewer commented:

P17, line 458: “...resulting in a final dataset of 630 unique crystal structures of drug-pocket complexes.” Although authors mention unique crystal structures, they fail to address the massive overrepresentation of certain drug-target families in the Protein Data Bank (PDB) (e.g., kinases, HIV protease, and heat shock proteins). Dozens or even hundreds of these PDBs could contain the exact same drug binding to the exact same protein pocket. The results will reflect the specific properties of a few over-studied protein families rather than the general characteristics of FDA-approved drug pockets.

Author reply:

We thank the reviewer for raising this important point. To address this comment, we analyzed the family-level composition of the FDA-AD dataset and included the results in Figure 5B. After applying the Tanimoto similarity = 1 filter described above, the dataset comprised 340 unique drug–pocket complexes spanning 31 distinct protein families. The most represented classes were transferases (20.8%), hydrolases (18.1%), oxidoreductases (14.8%), DNA-binding proteins (7.3%), and receptors (6.9%). This distribution suggests that the FDA-AD dataset is not dominated by a few over-studied protein families but instead reflects the general characteristics of drug-binding targets. These results and the corresponding discussion have been added to Section 2.5 of the revised manuscript as follows:

“Among the predicted human proteins, 17.9% were classified as transferases, 11.6% as hydrolases, and 9.4% each for DNA-binding proteins and oxidoreductases. In comparison, the 128 DEL target proteins were predominantly transferases (27.1%), followed by hydrolases (17.4%) and receptors (9.7%). The FDA-AD dataset, which encompasses 31 protein families, showed a similarly broad distribution, with transferases (20.8%), hydrolases (18.1%), oxidoreductases (14.8%), DNA-binding proteins (7.3%), and receptors (6.9%) as the most frequent classes. ... Interestingly, transferases, hydrolases, and oxidoreductases are consistently enriched across all three datasets (DEL, FDA-AD, and predicted DEL-compatible pockets), underscoring their central roles as preferred families for DEL screening.” (Section 2.5)

Response to Reviewer #3

1. The Reviewer commented:

Author reply:

We sincerely thank the reviewer who co-reviewed this manuscript. We truly appreciate their efforts and constructive feedback, which have contributed meaningfully to improving the clarity and rigor of our work.